# Noisy Interpolation Learning with Shallow Univariate ReLU Networks

**Nirmit Joshi**
TTI-Chicago
nirmit@ttic.edu

**Gal Vardi**
TTI-Chicago and Hebrew University
galvardi@ttic.edu

**Nathan Srebro**
TTI-Chicago
nati@ttic.edu

## Abstract

Understanding how overparameterized neural networks generalize despite perfect interpolation of noisy training data is a fundamental question. Mallinar et al. (2022) noted that neural networks seem to often exhibit "tempered overfitting", wherein the population risk does not converge to the Bayes optimal error, but neither does it approach infinity, yielding non-trivial generalization. However, this has not been studied rigorously. We provide the first rigorous analysis of the overfitting behavior of regression with minimum norm ($\ell_2$ of weights), focusing on univariate two-layer ReLU networks. We show overfitting is tempered (with high probability) when measured with respect to the $L_1$ loss, but also show that the situation is more complex than suggested by Mallinar et al., and overfitting is catastrophic with respect to the $L_2$ loss, or when taking an expectation over the training set.

## 1 Introduction

A recent realization is that, although sometimes overfitting can be catastrophic as suggested by our classic learning theory understanding, in other cases overfitting may not be so catastrophic. In fact, even *interpolation learning*, which entails achieving zero training error with noisy data, can still allow for good generalization, and even consistency (Zhang et al., 2017; Belkin et al., 2018). This has led to efforts towards understanding the nature of overfitting: how *benign* or *catastrophic* it is, and what determines this behavior, in different settings and using different models.

Although interest in benign overfitting stems from the empirical success of interpolating large neural networks, theoretical study so far has been mostly limited to linear and kernel methods, or to classification settings where the data is already linearly separable, with very high data dimension (tending to infinity as the sample size grows)[1]. But what about noisy interpolation learning in low dimensions, using neural networks?

---

[1] Minimum $\ell_2$ norm linear prediction (aka ridgeless regression) with noisy labels and (sub-)Gaussian features has been studied extensively (e.g. Hastie et al., 2020; Belkin et al., 2020; Bartlett et al., 2020; Muthukumar et al., 2020; Negrea et al., 2020; Chinot & Lerasle, 2020; Koehler et al., 2021; Wu & Xu, 2020; Tsigler & Bartlett, 2020; Zhou et al., 2022; Wang et al., 2022; Chatterji et al., 2021; Bartlett & Long, 2021; Shamir, 2022; Ghosh & Belkin, 2022; Chatterji & Long, 2021; Wang & Thrampoulidis, 2021; Cao et al., 2021; Muthukumar et al., 2021; Montanari et al., 2020; Liang & Recht, 2021; Thrampoulidis et al., 2020; Wang et al., 2021; Donhauser et al., 2022; Frei et al., 2023), and noisy minimum $\ell_1$ linear prediction (aka Basis Persuit) has also been considered (e.g. Ju et al., 2020; Koehler et al., 2021; Wang et al., 2022). Either way, these analyses are all in the high dimensional setting, with dimension going to infinity, since to allow for interpolation the dimension must be high, higher than the number of samples. Kernel methods amount to a minimum $\ell_2$ norm linear prediction, with very non-Gaussian features. But existing analyses of interpolation learning with kernel methods rely on "Gaussian Universality": either assuming as an ansatz the behavior is as for Gaussian features (Mallinar et al., 2022) or establishing this rigorously in certain high dimensional scalings (Hastie et al., 2019; Misiakiewicz, 2022; Mei & Montanari, 2022). In particular, such analyses are only valid when the input dimension goes to infinity (though possibly slower than the number of samples) and not for fixed low or moderate dimensions. Frei et al. (2022; 2023); Cao et al. (2022); Kou et al. (2023) study interpolation learning with neural networks, but only with high input dimension and when the data is interpolatable also with a linear predictor—in these cases, although non-linear neural networks are used, the results show they behave similarly to linear predictors. Manoj & Srebro (2023) take the other extreme and study interpolation learning with "short programs", which are certainly non-linear, but this is an abstract model that does not directly capture learning with neural networks.

Mallinar et al. (2022) conducted simulations with neural networks and observed *"tempered"* overfitting: the asymptotic risk does not approach the Bayes-optimal risk (there is no consistency), but neither does it diverge to infinity catastrophically. Such "tempered" behavior is well understood for 1-nearest neighbor, where the asymptotic risk is roughly twice the Bayes risk (Cover & Hart, 1967), and Mallinar et al. heuristically explain it also for some kernel methods. However, we do not have a satisfying and rigorous understanding of such behavior in neural networks, nor a more quantitative understanding of just how bad the risk might be when interpolating noisy data using a neural net.

In this paper, we begin rigorously studying the effect of overfitting in the noisy regression setting, with neural networks in low dimensions, where the data is *not* linearly interpolatable. Specifically, we study interpolation learning of univariate data (i.e. in one dimension) using a two-layer ReLU network (with a skip connection), which is a predictor $f_{\theta,a_0,b_0} : \mathbb{R} \to \mathbb{R}$ given by:

$$f_{\theta,a_0,b_0}(x) = \sum_{j=1}^{m} a_j(w_j x + b_j)_+ + a_0 x + b_0 \,, \tag{1}$$

where $\theta \in \mathbb{R}^{3m}$ denotes the weights (parameters) $\{a_j, w_j, b_j\}_{j=1}^{m}$. To allow for interpolation we do not limit the width $m$, and learn by minimizing the norm of the weights (Savarese et al., 2019; Ergen & Pilanci, 2021; Hanin, 2021; Debarre et al., 2022; Boursier & Flammarion, 2023):

$$\hat{f}_S = \arg\min_{f_{\theta,a_0,b_0}} \|\theta\|^2 \;\; \text{s.t.} \;\; \forall i \in [n], \; f_{\theta,a_0,b_0}(x_i) = y_i \;\; \text{where} \; S = \{(x_1, y_1), \ldots, (x_n, y_n)\}. \tag{2}$$

Following Boursier & Flammarion (2023) we allow an unregularized skip-connection in equation 1, where the weights $a_0, b_0$ of this skip connection are not included in the norm $\|\theta\|$ in equation 2. This skip connection avoids some complications and allows better characterizing $\hat{f}_S$ but does not meaningfully change the behavior (see Section 2).

**Why min norm?** Using unbounded size minimum weight-norm networks is natural for interpolation learning. It parallels the study of minimum norm high (even infinite) dimension linear predictors. For interpolation, we must allow the number of parameters to increase as the sample size increases. But to have any hope of generalization, we must choose among the infinitely many zero training error networks somehow, and it seems that some sort of explicit or implicit low norm bias is the driving force in learning with large overparameterized neural networks (Neyshabur et al., 2014). Seeking minimum $\ell_2$ norm weights is natural, e.g. as a result of small weight decay. Even without explicit weight decay, optimizing using gradient descent is also related to an implicit bias toward low $\ell_2$ norm: this can be made precise for linear models and for classification with ReLU networks (Chizat & Bach, 2020; Safran et al., 2022). For regression with ReLU networks, as we study here, the implicit bias is not well understood (see Vardi (2023)), and studying equation 2 is a good starting point for understanding the behavior of networks learned via gradient descent even without explicit weight decay. Interestingly, minimum-norm interpolation corresponds to the *rich regime*, and does not correspond to any kernel (Savarese et al., 2019). For the aforementioned reasons, understanding the properties of min-norm interpolators has attracted much interest in recent years (Savarese et al., 2019; Ongie et al., 2019; Ergen & Pilanci, 2021; Hanin, 2021; Debarre et al., 2022; Boursier & Flammarion, 2023).

**Noisy interpolation learning.** We consider a noisy distribution $\mathcal{D}$ over $[0, 1] \times \mathbb{R}$:

$$x \sim \text{Uniform}([0, 1]) \quad \text{and} \quad y = f^*(x) + \epsilon \; \text{with} \; \epsilon \; \text{independent of} \; x, \tag{3}$$

where $x$ is uniform for simplicity and concreteness[2]. The noise $\epsilon$ follows some arbitrary (but non-zero) distribution, and learning is based on an i.i.d. training set $S \sim \mathcal{D}^n$. Since the noise is non-zero, the "ground truth" predictor $f^*$ has non-zero training error, seeking a training error much smaller than that of $f^*$ would be overfitting (fitting the noise) and necessarily cause the complexity (e.g. norm) of the learned predictor to explode. The "right" thing to do is to balance between the training error and the complexity $\|\theta\|$. Indeed, under mild assumptions, this balanced approach leads to asymptotic consistency, with $\hat{f}_S \xrightarrow{n \to \infty} f^*$ and the asymptotic population risk of $\hat{f}_S$ converging to the Bayes risk. But what happens when we overfit and use the interpolating learning rule equation 2?

---

[2]All our results should also hold for any absolutely continuous distribution with bounded density and support. Roughly speaking, this can be achieved by dividing the support into disjoint intervals such that the distribution in each interval is well-approximated by a uniform distribution.

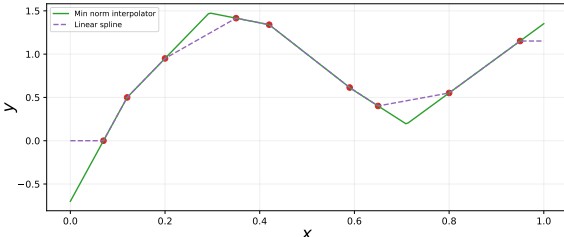

Figure 1: Comparison between linear-spline (purple) and min-norm (green) interpolators.

**Linear Splines.**   At first glance, we might be tempted to think that two-layer ReLUs behave like linear splines (see Figure 1). Indeed, if minimizing the norm of weights $w_i$ and $a_i$ but *not* the biases $b_i$ in equation 2, linear splines are a valid minimizer (Savarese et al., 2019; Ergen & Pilanci, 2021). As the number of noisy training points increases, linear splines "zig-zag" with tighter "zigs" but non-vanishing "amplitude" around $f^*$, resulting in an interpolator which roughly behaves like $f^*$ plus some added non-vanishing "noise". This does not lead to consistency, but is similar to a nearest-neighbor predictor (each prediction is a weighted average of two neighbors). Indeed, in Theorem 1 of Section 3, we show that linear splines exhibit "tempered" behavior, with asymptotic risk proportional to the noise level.

**From Splines to Min-Norm ReLU Nets.**   It turns out minimum norm ReLU networks, although piecewise linear, are not quite linear splines: roughly speaking, and as shown in Figure 1, they are more conservative in the number of linear "pieces". Because of this, in convex (conversely, concave) regions of the linear spline, minimum norm ReLU nets "overshoot" the linear spline in order to avoid breaking linear pieces. This creates additional "spikes", extending above and below the data points (see Figures 1 and 2) and thus potentially increasing the error. In fact, such spikes are also observed in interpolants reached by gradient descent (Shevchenko et al., 2022, Figure 1). How bad is the effect of such spikes on the population risk?

OUR CONTRIBUTION

**Effect of Overfitting on $L_p$ Risk.**   It turns out the answer is quite subtle and, despite considering the same interpolator, the nature of overfitting actually depends on how we measure the error. For a function $f : \mathbb{R} \to \mathbb{R}$, we measure its $L_p$ population error and the reconstruction error respectively as

$$\mathcal{L}_p(f) := \mathop{\mathbb{E}}_{(x,y)\sim\mathcal{D}}[|f(x) - y|^p] \quad \text{and} \quad \mathcal{R}_p(f) := \mathop{\mathbb{E}}_{x\sim\text{Uniform}([0,1])}[|f(x) - f^*(x)|^p].$$

We show in Theorems 2 and 3 of Section 4.2 that for $1 \le p < 2$,

$$\mathcal{L}_p(\hat{f}_S) \xrightarrow{n\to\infty} \Theta\left(\frac{1}{(2-p)_+}\right) \mathcal{L}_p(f^*). \tag{4}$$

This is an upper bound for any Lipschitz target $f^*$ and any noise distribution, and it is matched by a lower bound for Gaussian noise. That is, for abs-loss ($L_1$ risk), as well as any $L_p$ risk for $1 \le p < 2$, overfitting is **tempered**. But this tempered behavior explodes as $p \to 2$, and we see a sharp transition. We show in Theorem 4 of Section 4.3 that for any $p \ge 2$, including for the square loss ($p = 2$), in the presence of noise, $\mathcal{L}_p(\hat{f}_S) \xrightarrow{n\to\infty} \infty$ and overfitting is **catastrophic**.

**Convergence vs. Expectation.**   The behavior is even more subtle, in that even for $1 \le p < 2$, although the risk $\mathcal{L}_p(\hat{f}_S)$ converges in probability to a tempered behavior as in equation 4, its *expectation* is infinite: $\mathbb{E}_S[\mathcal{L}_p(\hat{f}_S)] = \infty$. Note that in studying tempered overfitting, Mallinar et al. (2022) focused on this expectation, and so would have categorized the behavior as "catastrophic" even for $p = 1$, emphasizing the need for more careful consideration of the effect of overfitting.

**I.I.D. Samples vs. Samples on a Grid.**   The catastrophic effect of interpolation on the $L_p$ risk with $p \ge 2$ is a result of the effect of fluctuations in the *spacing* of the training points. Large, catastrophic, spikes are formed by training points extremely close to their neighbors but with different labels (see Figures 2 and 5). To help understand this, in Section 5 we study a "fixed design" variant

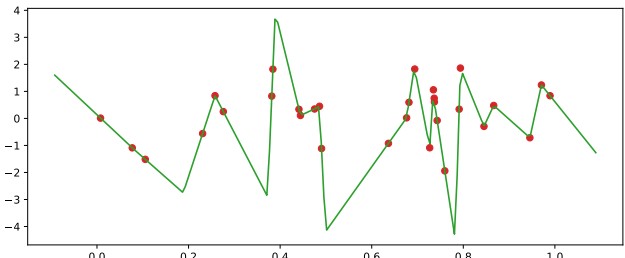

Figure 2: The min-norm interpolator for 30 random points with $f^* \equiv 0$ and $\mathcal{N}(0, 1)$ label noise.

of the problem, where the training inputs lie on a uniform grid, $x_i = i/n$, and responses follow $y_i = f^*(x_i) + \epsilon_i$. In this case, interpolation is always tempered, with $\mathcal{L}_p(\hat{f}_S) \xrightarrow{n \to \infty} O(\mathcal{L}_p(f^*))$ for any constant $p \geq 1$ (Theorem 5 of Section 5).

**Discussion and Takeaways.** Our work is the first to study noisy interpolation learning with min-norm ReLU networks for regression. It is also the first to study noisy interpolation learning in neural networks where the input dimension does not grow with the sample size, and to consider non-linearly-interpolatable data distributions (see below for a comparison with concurrent work in a classification setting). The univariate case might seem simplistic, but is a rich and well-studied model in its own right (Shevchenko et al., 2022; Ergen & Pilanci, 2021; Hanin, 2021; Debarre et al., 2022; Boursier & Flammarion, 2023; Williams et al., 2019; Mulayoff et al., 2021; Safran et al., 2022), and as we see, it already exhibits many complexities and subtleties that need to be resolved, and is thus a non-trivial necessary first step if we want to proceed to the multivariate case.

The main takeaway from our work is that the transition from tempered to catastrophic overfitting can be much more subtle than previously discussed, both in terms of the details of the setting (e.g., sampled data vs. data on a grid) and in terms of the definition and notion of overfitting (the loss function used, and expectation vs. high probability). Understanding these subtleties is crucial before moving on to more complex models.

More concretely, we see that for the square loss, the behavior does not fit the "tempered overfitting" predictions of Mallinar et al. (2022), and for the $L_1$ loss we get a tempered behavior with high probability but not in expectation, which highlights that the definitions of (Mallinar et al., 2022) need to be refined. We would of course not get such strange behavior with the traditional non-overfitting approach of balancing training error and norm; in this situation the risk converges almost surely to the optimal risk, with finite expectation and vanishing variances. Moreover, perhaps surprisingly, when the input data is on the grid (equally spaced), the behavior is tempered for all losses even in the presence of label noise. This demonstrates that the catastrophic behavior for $L_p$ losses for $p \geq 2$ is not just due to the presence of label noise; it is the combination of label noise and sampling of points that hurts generalization. We note that previous works considered benign overfitting with data on the grid as a simplified setting, which may help in understanding more general situations (Beaglehole et al., 2022; Lai et al., 2023). Our results imply that this simplification might change the behavior of the interpolator significantly. In summary, the nature of overfitting is a delicate property of the combination of how we measure the loss and how training examples are chosen.

**Comparison with concurrent work.** In a concurrent and independent work, Kornowski et al. (2023) studied interpolation learning in univariate two-layer ReLU networks in a classification setting, and showed that they exhibit tempered overfitting. In contrast to our regression setting, in classification only the output's sign affects generalization, and hence the height of the spikes do not play a significant role. As a result, our regression setting exhibits a fundamentally different behavior, and the above discussion on the delicateness of the overfitting behavior in regression does not apply to their classification setting.

## 2 REVIEW: MIN-NORM ReLU NETWORKS

Minimum-norm unbounded-width univariate two-layer ReLU networks have been extensively studied in recent years, starting with Savarese et al. (2019), with the exact formulation equation 2 in-

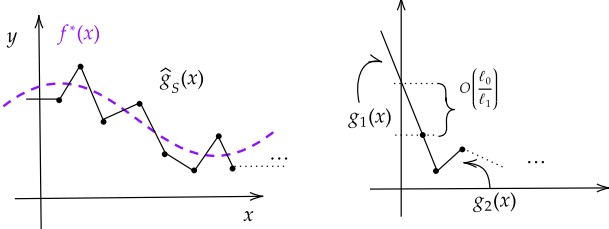

Figure 3: An illustration of the linear spline interpolator $\hat{g}_S$ (left), and of the variant $\hat{h}_S$ where linear pieces are extended beyond the endpoints (right).

corporating a skip connection due to Boursier & Flammarion (2023). Boursier & Flammarion, following prior work, establish that a minimum of equation 2 exists, with a finite number of units, and that it is also unique.

The problem in equation 2 is also equivalent to minimizing the "representation cost" $R(f) = \int_{\mathbb{R}} \sqrt{1+x^2}|f''(x)|dx$ over all interpolators $f$, although we will not use this characterization explicitly in our analysis. Compared to Savarese et al. (2019), where the representation cost is given by $\max\{\int |f''(x)|dx, |f'(-\infty) + f'(+\infty)|\}$, the weighting $\sqrt{1+x^2}$ is due to penalizing the biases $b_i$. More significantly, the skip connection in equation 1 avoids the "fallback" terms of $|f'(-\infty) + f'(+\infty)|$, which only kick-in in extreme cases (very few points or an extreme slope). This simplified the technical analysis and presentation, while rarely affecting the solution.

Boursier & Flammarion provide the following characterization of the minimizer[3] $\hat{f}_S$ of equation 2, which we will rely on heavily:

**Lemma 2.1** (Boursier & Flammarion (2023)). *For $0 \leq x_1 < x_2 < \cdots < x_n$, the problem in equation 2 admits a unique minimizer of the form:*

$$\hat{f}_S(x) = ax + b + \sum_{i=1}^{n-1} a_i(x - \tau_i)_+ , \qquad (5)$$

*where $\tau_i \in [x_i, x_{i+1})$ for every $i \in [n-1]$.*

As in the above characterization, it is very convenient to take the training points to be sorted. Since the learned network $\hat{f}_S$ does not depend on the order of the points, we can always "sort" the points without changing anything. And so, throughout the paper, we will always take the points to be sorted (formally, the results apply to i.i.d. points, and the analysis is done after sorting these points).

## 3 WARM UP: TEMPERED OVERFITTING IN LINEAR-SPLINE INTERPOLATION

We start by analyzing tempered overfitting for linear-spline interpolation. Namely, we consider the piecewise-linear function obtained by connecting each pair of consecutive points in the dataset $S \sim \mathcal{D}^n$ (see Figures 1 and 3 left) and analyze its test performance.

Given a dataset $S = \{(x_i, y_i)\}_{i=1}^{n}$, let $g_i : \mathbb{R} \to \mathbb{R}$ be the affine function joining the points $(x_i, y_i)$ and $(x_{i+1}, y_{i+1})$. Thus, $g_i$ is the straight line joining the endpoints of the $i$-th interval. Then, the linear spline interpolator $\hat{g}_S : [0, 1] \to \mathbb{R}$ is given by

$$\hat{g}_S(x) := y_1 \cdot \mathbf{1}\{x < x_1\} + y_n \cdot \mathbf{1}\{x \geq x_n\} + \sum_{i=1}^{n-1} g_i(x) \cdot \mathbf{1}\{x \in [x_i, x_{i+1})\}. \qquad (6)$$

---

[3]If the biases $b_i$ are *not* included in the norm $\|\theta\|$ in equation 2, and this norm is replaced with $\sum_i(a_i^2 + w_i^2)$, the modified problem admits multiple non-unique minimizers, including a linear spline (with modified behavior past the extreme points) (Savarese et al., 2019). This set of minimizers was characterized by Hanin (2021). Interestingly, the minimizer $\hat{f}_S$ of equation 2 (when the biases are included in the norm) is also a minimizer of the modified problem (without including the biases). All our results apply also to the setting without penalizing the biases in the following sense: the upper bounds are valid for all minimizers, while some minimizer, namely $\hat{f}_S$ that we study, exhibits the lower bound behavior.

Note that in the intervals $[0, x_1]$ and $[x_n, 1]$ the linear-spline $\hat{g}_S$ is defined to be constants that correspond to labels $y_1$ and $y_n$ respectively. The following theorem characterizes the asymptotic behavior of $\mathcal{L}_p(\hat{g}_S)$ for every $p \geq 1$:

**Theorem 1.** *Let $f^*$ be any Lipschitz function and $\mathcal{D}$ be the distribution from equation 3. Let $S \sim \mathcal{D}^n$, and $\hat{g}_S$ be the linear-spline interpolator (equation 6) w.r.t. the dataset $S$. Then, for any $p \geq 1$ there is a constant $C_p$ such that*

$$\lim_{n \to \infty} \mathbb{P}_S \left[ \mathcal{R}_p(\hat{g}_S) \leq C_p \, \mathcal{L}_p(f^*) \right] = 1 \quad and \quad \lim_{n \to \infty} \mathbb{P}_S \left[ \mathcal{L}_p(\hat{g}_S) \leq C_p \, \mathcal{L}_p(f^*) \right] = 1.$$

The theorem shows that the linear-spline interpolator exhibits tempered behavior, namely, w.h.p. over $S$ the interpolator $\hat{g}_S$ performs like the predictor $f^*$, up to a constant factor. To understand why Theorem 1 holds, note that for all $i \in [n-1]$ and $x \in [x_i, x_{i+1}]$ the linear-spline interpolator satisfies $\hat{g}_S(x) \in [\min\{y_i, y_{i+1}\}, \max\{y_i, y_{i+1}\}]$. Moreover, we have for all $i \in [n]$ that $|y_i - f^*(x_i)| = |\epsilon_i|$, where $\epsilon_i$ is the random noise. Using these facts, it is not hard to bound the expected population loss of $\hat{g}_S$ in each interval $[x_i, x_{i+1}]$, and by using the law of large numbers it is also possible to bound the probability (over $S$) that the loss in the domain $[0, 1]$ is large. Thus, we can bound the $L_p$ loss both in expectation and in probability.

**Delicate behavior of linear splines.** We now consider the following variant of the linear-spline interpolator:

$$\hat{h}_S(x) := g_1(x) \cdot \mathbf{1}\{x < x_1\} + g_{n-1}(x) \cdot \mathbf{1}\{x > x_n\} + \hat{g}_S(x) \cdot \mathbf{1}\{x \in [x_1, x_n]\}. \tag{7}$$

In words, $\hat{h}_S$ is exactly the same as $\hat{g}_S$ in the interval $[x_1, x_n]$, but it extends the linear pieces $g_1$ and $g_{n-1}$ beyond the endpoints $x_1$ and $x_n$ (respectively), as illustrated in Figure 3 (right). The interpolator $\hat{h}_S$ still exhibits tempered behavior in probability, similarly to $\hat{g}_S$. However, perhaps surprisingly, $\hat{h}_S$ is not tempered in expectation (see Appendix A for details). This delicate behavior of the linear-spline interpolator is important since in the next section we will show that the min-norm interpolator has a similar behavior to $\hat{h}_S$ in the intervals $[0, x_1], [x_n, 1]$, and as a consequence, it is tempered with high probability but not in expectation.

## 4 MIN-NORM INTERPOLATION WITH RANDOM DATA

In this section, we study the performance of the min-norm interpolator with random data. We first present some important properties of the min-norm interpolator in Section 4.1. In Sections 4.2 and 4.3 we use this characterization to study its performance.

### 4.1 CHARACTERIZING THE MIN-NORM INTERPOLATOR

Our goal is to give a characterization of the min-norm interpolator $\hat{f}_S(x)$ (equation 5), in terms of linear splines as defined in equation 6. Recall the definition of affine functions $g_1(x), \ldots, g_{n-1}(x)$, which are piece-wise affine functions joining consecutive points. Let $\delta_i$ be the slope of the line $g_i(x)$, i.e. $\delta_i = g_i'(x)$. We denote $\delta_0 := \delta_1$ and $\delta_n := \delta_{n-1}$. Then, we can define the sign of the curvature of the linear spline $\hat{g}_S(x)$ at each point.

**Definition 4.1.** *For any $i \in [n]$,*

$$\mathsf{curv}(x_i) = \begin{cases} +1 & \delta_i > \delta_{i-1} \\ 0 & \delta_i = \delta_{i+1} \\ -1 & \delta_i < \delta_{i-1} \end{cases}$$

Based on the curvature, the following lemma geometrically characterizes $\hat{f}_S$ in any interval $[x_i, x_{i+1})$, in terms of the linear pieces $g_{i-1}, g_i, g_{i+1}$.

**Lemma 4.2.** *The function $\hat{f}_S$ can be characterized as follows:*

- $\hat{f}_S(x) = g_1(x)$ *for $x \in (-\infty, x_2)$;*

- $\hat{f}_S(x) = g_{n-1}(x)$ *for $x \in [x_{n-1}, \infty)$;*

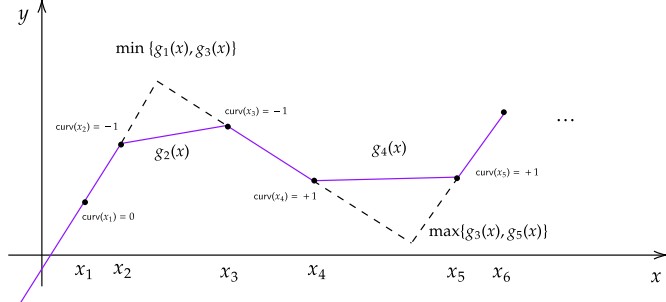

Figure 4: An illustration of the characterization of $\hat{f}_S$ from Lemma 4.2.

- *In each interval $[x_i, x_{i+1})$ for $i \in \{2, \ldots n-2\}$,*

    1. *If $\mathsf{curv}(x_i) = \mathsf{curv}(x_{i+1}) = +1$ then*
    $$\max\{g_{i-1}(x), g_{i+1}(x)\} \leq \hat{f}_S(x) \leq g_i(x);$$

    2. *If $\mathsf{curv}(x_i) = \mathsf{curv}(x_{i+1}) = -1$ then*
    $$\min\{g_{i-1}(x), g_{i+1}(x)\} \geq \hat{f}_S(x) \geq g_i(x);$$

    3. *Else, i.e. either $\mathsf{curv}(x_i) = 0$ or $\mathsf{curv}(x_{i+1}) = 0$ or $\mathsf{curv}(x_i) \neq \mathsf{curv}(x_{i+1})$,*
    $$\hat{f}_S(x) = g_i(x).$$

The lemma implies that $\hat{f}_S$ coincides with $\hat{g}_S$ except in an interval $[x_i, x_{i+1})$ where the curvature of the two points are both $+1$ or $-1$ (see Figure 4). Intuitively, this property captures the worst-case effect of the spikes and will be crucial in showing the tempered behavior of $\hat{f}_S$ w.r.t. $L_p$ for $p \in [1, 2)$. However, this still does not imply that such spikes are necessarily formed.

To this end, Boursier & Flammarion (2023, Lemma 8) characterized the situation under which indeed these spikes are formed. Roughly speaking, if the sign of the curvature changes twice within three points, then we get a spike. Formally, we identify special points from left to right recursively where the sign of the curvature changes.

**Definition 4.3.** *We define $n_1 := 1$. Having defined the location of the special points $n_1, \ldots, n_{i-1}$, we recursively define*
$$n_i = \min\{j > n_{i-1} : \mathsf{curv}(x_j) \neq \mathsf{curv}(x_{n_i})\}.$$
*If there is no such $n_{i-1} < j \leq n$ where $\mathsf{curv}(x_j) \neq \mathsf{curv}(x_{n_i})$, then $n_{i-1}$ is the location of the last special point.*

**Lemma 4.4** (Boursier & Flammarion (2023)). *For any $k \geq 1$, if $\delta_{n_k-1} \neq \delta_{n_k}$ and $n_{k+1} = n_k + 2$, then $\hat{f}_S$ has exactly one kink between $(x_{n_k-1}, x_{n_k+1})$. Moreover, if $\mathsf{curv}(x_{n_k}) = \mathsf{curv}(x_{n_k+1}) = -1$ then $\hat{f}_S(x) = \min\{g_{n_k-1}(x), g_{n_k+1}(x)\}$ in $[x_{n_k}, x_{n_k+1}]$.*

This is a slight variation of (Boursier & Flammarion, 2023, Lemma 8), which we reprove in the appendix for completeness. See Figure 5 for an illustration of the above lemma. To show the catastrophic behavior of $\hat{f}_S$ for $p \geq 2$, we will consider events under which such configurations of points are formed. This will result in spikes giving catastrophic behavior.

### 4.2 TEMPERED OVERFITTING FOR $L_p$ WITH $p \in [1, 2)$

We now show the tempered behavior of the minimal norm interpolator w.r.t. $L_p$ losses for $p \in [1, 2)$.

**Theorem 2.** *Let $f^*$ be a Lipschitz function and $\mathcal{D}$ be the distribution from equation 3. Sample $S \sim \mathcal{D}^n$, and let $\hat{f}_S$ be the min-norm interpolator (equation 5). Then, for some universal constant $C > 0$, for any $p \in [1, 2)$ we have*

$$\lim_{n \to \infty} \mathbb{P}_S\left[\mathcal{R}_p(\hat{f}_S) \leq \frac{C}{2-p} \cdot \mathcal{L}_p(f^*)\right] = 1 \quad and \quad \lim_{n \to \infty} \mathbb{P}_S\left[\mathcal{L}_p(\hat{f}_S) \leq \frac{C}{2-p} \cdot \mathcal{L}_p(f^*)\right] = 1.$$

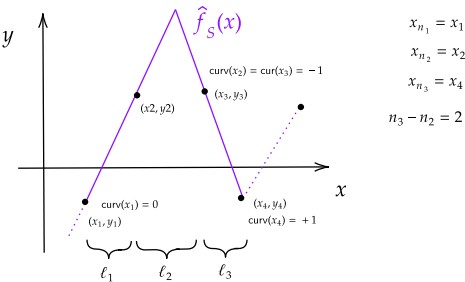

Figure 5: An illustration of the spike formed by Lemma 4.4. Here, $x_2$ and $x_4$ are two consecutive special points with exactly one point in between. There must be exactly one kink in $(x_1, x_4)$. Thus, in $[x_2, x_3)$, the interpolator $\hat{f}_S$ must be $\min\{g_1(x), g_3(x)\}$.

The proof of Theorem 2 builds on Lemma 4.2, which implies that in an interval $[x_i, x_{i+1})$, a spike in the interpolator $\hat{f}_S$ must be bounded within the triangle obtained from $g_{i-1}, g_i, g_{i+1}$ (see Figure 4). Analyzing the population loss of $\hat{f}_S$ requires considering the distribution of the spacings between data points. Let $\ell_0, \ldots, \ell_n$ be such that

$$\forall i \in [n-1] \quad \ell_i = x_{i+1} - x_i, \quad \ell_0 = x_1, \quad \ell_n = 1 - x_n \, . \tag{8}$$

Prior works (Alagar, 1976; Pinelis, 2019) established that

$$(\ell_0, \ldots, \ell_n) \sim \left( \frac{X_0}{X}, \ldots, \frac{X_n}{X} \right), \text{ where } X_0, \ldots, X_n \stackrel{\text{i.i.d.}}{\sim} \text{Exp}(1), \text{ and } X := \sum_{i=0}^{n} X_i \, . \tag{9}$$

The slopes of the affine functions $g_{i-1}, g_{i+1}$ are roughly $\frac{1}{\ell_{i-1}}, \frac{1}{\ell_{i+1}}$, where $\ell_j$ are the lengths as defined in equation 8. Hence, the spike's height is proportional to $\frac{\ell_i}{\max\{\ell_{i-1}, \ell_{i+1}\}}$. As a result, the $L_p$ loss in the interval $[x_i, x_{i+1}]$ is roughly

$$\left( \frac{\ell_i}{\max\{\ell_{i-1}, \ell_{i+1}\}} \right)^p \cdot \ell_i = \frac{\ell_i^{p+1}}{\max\{\ell_{i-1}, \ell_{i+1}\}^p} \, .$$

Using the distribution of the $\ell_j$'s given in equation 9, we can bound the expectation of this expression. Then, similarly to our discussion on linear splines in Section 3, in the range $[x_1, x_n]$ we can bound the $L_p$ loss both in expectation and in probability. In the intervals $[0, x_1]$ and $[x_n, 1]$, the expected loss is infinite (similarly to the interpolator $\hat{h}_S$ in equation 7), and therefore we have

$$\mathbb{E}_S \left[ \mathcal{L}_p(\hat{f}_S) \right] = \infty \, . \tag{10}$$

Still, we can get a high probability upper bound for the $L_p$ loss in the intervals $[0, x_1]$ and $[x_n, 1]$. Thus, we get a bound on $L_p$ loss in the entire domain $[0, 1]$ w.h.p. We note that the definition of tempered overfitting in Mallinar et al. (2022) considers only the expectation. Theorem 2 and equation 10 imply that in our setting we have tempered behavior in probability but not in expectation, which demonstrates that tempered behavior is delicate.

We also show a lower bound for the population loss $L_p$ which matches the upper bound from Theorem 2 (up to a constant factor independent of $p$). The lower bound holds already for $f^* \equiv 0$ and Gaussian label noise.

**Theorem 3.** *Let $f^* \equiv 0$, consider label noise $\epsilon \sim \mathcal{N}(0, \sigma^2)$ for some constant $\sigma > 0$, and let $\mathcal{D}$ be the corresponding distribution from equation 3. Let $S \sim \mathcal{D}^n$, and let $\hat{f}_S$ be the min-norm interpolator (equation 5). Then, for some universal constant $c > 0$, for any $p \in [1, 2)$ we have*

$$\lim_{n \to \infty} \mathbb{P}_S \left[ \mathcal{R}_p(\hat{f}_S) \geq \frac{c}{2-p} \cdot \mathcal{L}_p(f^*) \right] = 1 \quad and \quad \lim_{n \to \infty} \mathbb{P}_S \left[ \mathcal{L}_p(\hat{f}_S) \geq \frac{c}{2-p} \cdot \mathcal{L}_p(f^*) \right] = 1 \, .$$

The proof of the above lower bound follows similar arguments to the proof of catastrophic overfitting for $p \geq 2$, which we will discuss in the next section.

### 4.3 CATASTROPHIC OVERFITTING FOR $L_p$ WITH $p \geq 2$

Next, we prove that for the $L_p$ loss with $p \geq 2$, the min-norm interpolator exhibits catastrophic overfitting. We prove this result already for $f^* \equiv 0$ and Gaussian label noise:

**Theorem 4.** *Let $f^* \equiv 0$, consider label noise $\epsilon \sim \mathcal{N}(0, \sigma^2)$ for some constant $\sigma > 0$, and let $\mathcal{D}$ be the corresponding distribution from equation 3. Let $S \sim \mathcal{D}^n$, and let $\hat{f}_S$ be the min-norm interpolator (equation 5). Then, for any $p \geq 2$ and $b > 0$,*

$$\lim_{n \to \infty} \mathbb{P}_S \left[ \mathcal{R}_p(\hat{f}_S) > b \right] = 1 \quad and \quad \lim_{n \to \infty} \mathbb{P}_S \left[ \mathcal{L}_p(\hat{f}_S) > b \right] = 1 \,.$$

To obtain some intuition on this phenomenon, consider the first four samples $(x_1, y_1), \ldots, (x_4, y_4)$, and let $\ell_i$ be the lengths of the intervals as defined in equation 8. We show that with constant probability, the configuration of the labels of these samples satisfies certain properties, which are illustrated in Figure 5. In this case, Lemma 4.4 implies that in the interval $[x_2, x_3]$ the interpolator $\hat{f}_S$ is equal to $\min\{g_1(x), g_3(x)\}$, where $g_1$ (respectively, $g_3$) is the affine function that connects $x_1, x_2$ (respectively, $x_3, x_4$). Now, as can be seen in the figure, in this "unfortunate configuration" the interpolator $\hat{f}_S$ spikes above $f^* \equiv 0$ in the interval $[x_2, x_3]$, and the spike's height is proportional to $\frac{\ell_2}{\max\{\ell_1, \ell_3\}}$. As a result, the $L_p$ loss in the interval $[x_2, x_3]$ is roughly $\frac{\ell_2^{p+1}}{\max\{\ell_1, \ell_3\}^p}$. Using equation 9, we can show that $\mathbb{E}_S \left[ \frac{\ell_2^{p+1}}{\max\{\ell_1, \ell_3\}^p} \right] = \infty$ for any $p \geq 2$.

We then divide divide the $n$ samples in $S$ into $\Theta(n)$ disjoint subsets and consider the events that labels are such that the 4 middle points exhibit an "unfortunate configuration" as described above. Using the fact that we have $\Theta(n)$ such subsets and the losses in these subsets are only mildly correlated, we are able to prove that $\hat{f}_S$ exhibits a catastrophic behavior also in probability.

We note that the proof of Theorem 3 follows similar arguments, except that when $p < 2$ the expectation of the $L_p$ loss in each subset with an "unfortunate configuration" is finite, and hence we get a finite lower bound.

## 5 MIN-NORM INTERPOLATION WITH SAMPLES ON THE GRID

In this section, we analyze the population loss of the min-norm interpolator, when the $n$ data-points in $S$ are uniformly spaced, instead of i.i.d. uniform sampling considered in the previous sections. Namely, consider the training set $S = \{(x_i, y_i) : i \in [n]\}$, where

$$x_i = \frac{i}{n} \quad \text{and} \quad y_i = f^*(x_i) + \epsilon_i \quad \text{for i.i.d. noise } \epsilon_i \,. \tag{11}$$

Note that the randomness in $S$ is only in the label noises $\epsilon_i$. It can be interpreted as a *non-adaptive active learning* setting, where the learner can actively choose the training points, and then observe noisy measurements at these points, and the query points are selected on an equally spaced grid. We show that in this situation the min-norm interpolator exhibits tempered overfitting with respect to any $L_p$ loss:

**Theorem 5.** *Let $f^*$ be any Lipschitz function. For the size-$n$ dataset $S$ given by equation 11, let $\hat{f}_S$ be the min-norm interpolator (equation 5). Then for any $p \geq 1$, there is a constant $C_p$ such that*

$$\lim_{n \to \infty} \mathbb{P}_S \left[ \mathcal{R}_p(\hat{f}_S) \leq C_p \, \mathcal{L}_p(f^*) \right] = 1 \quad and \quad \lim_{n \to \infty} \mathbb{P}_S \left[ \mathcal{L}_p(\hat{f}_S) \leq C_p \, \mathcal{L}_p(f^*) \right] = 1 \,.$$

An intuitive explanation is as follows. Since the points are uniformly spaced, whenever spikes are formed, they can at most reach double the height without the spikes. Thus, the population loss of $\hat{f}_S(x)$ becomes worse but only by a constant factor. We remark that in this setting the min-norm interpolator exhibits tempered overfitting both in probability (as stated in Theorem 5) and in expectation. From Theorem 5 we conclude that the catastrophic behavior for $L_p$ with $p \geq 2$ shown in Theorem 4 stems from the non-uniformity in the lengths of the intervals $[x_i, x_{i+1}]$, which occurs when the $x_i$'s are drawn at random.

ACKNOWLEDGEMENTS

This research was done as part of the NSF-Simons Sponsored Collaboration on the Theoretical Foundations of Deep Learning and the NSF Tripod Institute on Data, Econometrics, Algorithms, and Learning (IDEAL). N. J. would like to thank Surya Pratap Singh for his generous time in helping resolve Python errors.

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

## A    DELICATE BEHAVIOR OF LINEAR SPLINES

Recall the definition of $\hat{h}_S$ from equation 7.

$$\hat{h}_S(x) := g_1(x) \cdot \mathbf{1}\{x < x_1\} + g_{n-1}(x) \cdot \mathbf{1}\{x > x_n\} + \hat{g}_S(x) \cdot \mathbf{1}\{x \in [x_1, x_n]\}.$$

For simplicity, assume that $f^* \equiv 0$ and consider the $L_1$ loss. Since in the interval $[0, x_1]$ the interpolator $\hat{h}_S$ is defined by extending the line connecting $(x_1, y_1)$ and $(x_2, y_2)$, then it has slope of $\Theta\left(\frac{1}{\ell_1}\right)$, and hence the $L_1$ loss of $\hat{h}_S$ in $[0, x_1]$ is

$$\Theta\left(\frac{\ell_0^2}{\ell_1}\right) = \Theta\left(\frac{X_0^2}{X \cdot X_1}\right),$$

as can be seen in Figure 3 (right). Recall that $(X_0, \ldots, X_n)$ and $X$ are defined in equation 9. Since $X_1 \sim \text{Exp}(1)$, then $\mathbb{E}\left[\frac{1}{X_1}\right] = \infty$, and as a consequence the expected $L_1$ loss in $[0, x_1]$ is infinite. A similar argument also holds for the interval $[x_n, 1]$. Thus, we get that $\mathbb{E}_S\left[\mathcal{L}_p(\hat{h}_S)\right] = \infty$. However, with high probability the lengths $\ell_1$ and $\ell_{n-1}$ will not be too short, and therefore the loss in the intervals $[0, x_1]$ and $[x_n, 1]$ will be bounded, which implies tempered overfitting with high probability.

## B    PROOF OF THEOREM 1

*Proof of Theorem 1.* Let $G < \infty$ be the Lipschitz constant of $f^*$. We sample $S \sim \mathcal{D}^n$ and we number points $(x_i, y_i)$ such that:

$$0 < x_1 < x_2 < \cdots < x_n < 1.$$

We will also denote $x_0 = 0$ and $x_{n+1} = 1$ for simplicity of exposition. Our goal is to analyze the population and reconstruction errors of linear splines $\hat{g}_S(x)$, as defined in equation 6.

$$\begin{aligned}
\mathcal{L}_p(\hat{g}_S) &= \mathop{\mathbb{E}}_{(x,y)\sim\mathcal{D}}\left[|\hat{g}_S(x) - y|^p\right] \\
&= \mathop{\mathbb{E}}_{x\sim\text{Uniform}([0,1]),\epsilon}\left[|\hat{g}_S(x) - f^*(x) - \epsilon|^p\right] \\
&\leq 2^{p-1} \mathop{\mathbb{E}}_{x\sim\text{Uniform}([0,1]),\epsilon}\left[|\hat{g}_S(x) - f^*(x)|^p + |\epsilon|^p\right] \\
&= 2^{p-1}\left(\mathop{\mathbb{E}}_{x\sim\text{Uniform}([0,1])}\left[|\hat{g}_S(x) - f^*(x)|^p\right] + \mathop{\mathbb{E}}_{\epsilon}\left[|\epsilon|^p\right]\right) \\
&= 2^{p-1}(\mathcal{R}_p(\hat{g}_S) + \mathcal{L}_p(f^*)) .
\end{aligned} \tag{12}$$

Therefore, it boils down to analyzing $\mathcal{R}_p(\hat{g}_S)$. We define the risk in the interval $[x_i, x_{i+1}]$ as the random variable $R_i$. In particular, for $i \in \{0, 1, \ldots, n\}$ as

$$R_i := \int_{x_i}^{x_{i+1}} |\hat{g}_S(x) - f^*(x)|^p \, dx, \quad \text{and} \quad \mathcal{R}_p(\hat{g}_S(x)) = \sum_{i=0}^{n} R_i. \tag{13}$$

The entire range from $[0, 1]$ is divided into $n + 1$ intervals. We denote their length by $\ell_0, \ldots, \ell_n$. In particular, $\ell_i := x_{i+1} - x_i$. Recall the joint distribution of $(\ell_0, \ldots, \ell_n)$ in equation 9.

We first show that the sum of the risks in the first and the last intervals is vanishing as $n \to \infty$.

**Lemma B.1.** *For any $\gamma > 0$, we have $\lim_{n\to\infty} \mathbb{P}_S[R_0 + R_n \leq \gamma] = 1$.*

All the helper lemmas, including the above, are proved at the end of the proof of the theorem. We now focus on bounding the remaining $R_i$'s. Define

$$R := \sum_{i=1}^{n-1} R_i, \tag{14}$$

then we are interested in bounding $R$. For any $i \in [n-1]$ and $x \in [x_i, x_{i+1}]$,

$$
\begin{aligned}
|\hat{g}_S(x) - f^*(x)| &= |g_i(x) - f^*(x)| \\
&= \left| y_i + \frac{(y_{i+1} - y_i)}{(x_{i+1} - x_i)}(x - x_i) - f^*(x) \right| \\
&= \left| f^*(x_i) + \epsilon_i + \left( \frac{f^*(x_{i+1}) + \epsilon_{i+1} - f^*(x_i) - \epsilon_i}{x_{i+1} - x_i} \right)(x - x_i) - f^*(x) \right| \\
&\leq |f^*(x_i) - f^*(x)| + |\epsilon_i| + \frac{G|x_{i+1} - x_i| + |\epsilon_{i+1} - \epsilon_i|}{x_{i+1} - x_i}(x - x_i) \\
&\leq G(x - x_i) + |\epsilon_i| + \left( \frac{G(x_{i+1} - x_i) + |\epsilon_{i+1}| + |\epsilon_i|}{x_{i+1} - x_i} \right)(x - x_i) \\
&\leq G \cdot \ell_i + |\epsilon_i| + \left( \frac{G \cdot \ell_i + |\epsilon_{i+1}| + |\epsilon_i|}{\ell_i} \right) \cdot \ell_i \\
&= 2G \cdot \ell_i + |\epsilon_{i+1}| + 2|\epsilon_i|.
\end{aligned}
$$

Therefore, for $i \in [n-1]$

$$
\begin{aligned}
R_i = \int_{x_i}^{x_{i+1}} |\hat{g}_S(x) - f^*(x)|^p \, dx &\leq \ell_i (2G \cdot \ell_i + |\epsilon_{i+1}| + 2|\epsilon_i|)^p \\
&\leq 3^{p-1} \ell_i ((2G)^p \ell_i^p + |\epsilon_{i+1}|^p + 2^p |\epsilon_i|^p) \\
&\leq 3^{p-1}(2G)^p \ell_i^{p+1} + 3^{p-1} \ell_i |\epsilon_{i+1}|^p + 3^{p-1} 2^p \ell_i |\epsilon_i|^p := \hat{R}_i
\end{aligned}
$$

Therefore, if we define $\hat{R} := \sum_{i=1}^{n-1} \hat{R}_i$ then it serves as upper bound for $R$, i.e. $R \leq \hat{R}$. Since the $\ell_i$'s are mildly dependent random variables; we will try to re-express $\hat{R}$ as the sum of independent random variables. We now define random variables $\tilde{\ell}_0, \ldots, \tilde{\ell}_n \overset{\text{i.i.d.}}{\sim} \text{Exp}(1)/(n+1)$. More specifically, $(\tilde{\ell}_0, \ldots, \tilde{\ell}_n) = (X_0, \ldots, X_n)/(n+1)$. Using these random variables, define random variables similar to $\hat{R}_1, \ldots, \hat{R}_{n-1}$, but replace $\ell_i$ with $\tilde{\ell}_i$

$$
\tilde{R}_i := 3^{p-1}(2G)^p \tilde{\ell}_i^{p+1} + 3^{p-1} \tilde{\ell}_i |\epsilon_{i+1}|^p + 3^{p-1} \cdot 2^p \tilde{\ell}_i |\epsilon_i|^p, \quad \text{and} \quad \tilde{R} = \sum_{i=1}^{n-1} \tilde{R}_i.
$$

Then, the following lemma (whose proof we include after the proof of the theorem) establishes the almost sure convergence between $\hat{R}$ and $\tilde{R}$. Therefore, it suffices to bound the latter.

**Lemma B.2.** *As $n \to \infty$, we have $\tilde{R} - \hat{R} \xrightarrow{\text{a.s.}} 0$.*

Still on looking at $\tilde{R}$, any two consecutive $\tilde{R}_i$ and $\tilde{R}_{i+1}$ are dependent since it shares $\epsilon_{i+1}$ in its definition. Due to this, we split $\tilde{R}$ into two sums $\tilde{R}_{\text{odd}}$ (and $\tilde{R}_{\text{even}}$) containing odd-numbered terms (and even-numbered terms respectively).

$$
\tilde{R}_{\text{odd}} := \sum_{i \in [n-1], i\%2=1} \tilde{R}_i, \quad \tilde{R}_{\text{even}} := \sum_{i \in [n-1], i\%2=0} \tilde{R}_i, \quad \text{and} \quad \tilde{R} = \tilde{R}_{\text{odd}} + \tilde{R}_{\text{even}}
$$

Now, $\tilde{R}_{\text{odd}}$ is the sum of $\lceil (n-1)/2 \rceil$ i.i.d. random variables. Similarly, $\tilde{R}_{\text{even}}$ is the sum of $\lfloor (n-1)/2 \rfloor$ i.i.d. random variables. Let us calculate the expectation of these identically distributed random variables, which are $\tilde{R}_i$'s. For any $i \in [n-1]$, Further simplifying:

$$
\tilde{R}_{\text{odd}} = \sum_{i \in [n-1], i\%2=1} \left( \frac{3^{p-1}(2G)^p X_i^{p+1}}{(n+1)^{p+1}} + \frac{3^{p-1}|\epsilon_{i+1}|^p X_i}{(n+1)} + \frac{3^{p-1} \cdot 2^p \cdot |\epsilon_i|^p X_i}{(n+1)} \right) \tag{15}
$$

By the strong law of large numbers (LLN), we can say that as $n \to \infty$

$$
\frac{1}{\lceil (n-1)/2 \rceil} \sum_{i \in [n-1], i\%2=1} X_i^{p+1} \xrightarrow{\text{a.s.}} \mathbb{E}[\text{Exp}(1)^{p+1}] = \Gamma(p+2)
$$

Therefore, as $n \to \infty$, for $p \geq 1$ the first term of equation 15

$$\sum_{i \in [n-1], i\%2=1} \frac{3^{p-1}(2G)^p X_i^{p+1}}{(n+1)^{p+1}} \xrightarrow{\text{a.s.}} 0. \tag{16}$$

Similarly, using the strong LLN

$$\frac{1}{\lceil (n-1)/2 \rceil} \sum_{i \in [n-1], i\%2=1} 3^{p-1}|\epsilon_{i+1}|^p X_i + 3^{p-1} \cdot 2^p |\epsilon_i|^p X_i \xrightarrow{\text{a.s.}} 3^{p-1} \, \mathbb{E} \left[ |\epsilon_{i+1}|^p X_i \right] + 3^{p-1} \cdot 2^p \, \mathbb{E}[|\epsilon_i|^p X_i],$$

$$= 3^{p-1} \mathcal{L}_p(f^*) + 3^{p-1} \cdot 2^p \mathcal{L}_p(f^*).$$

Therefore, as $n \to \infty$, the second and third terms of equation 15 converge almost surely as follows.

$$\sum_{i \in [n-1], i\%2=1} \left( \frac{3^{p-1}|\epsilon_{i+1}|^p X_i}{(n+1)} + \frac{3^{p-1} \cdot 2^p \cdot |\epsilon_i|^p X_i}{(n+1)} \right) \xrightarrow{\text{a.s.}} \frac{3^{p-1} \mathcal{L}_p(f^*) + 3^{p-1} \cdot 2^p \mathcal{L}_p(f^*)}{2}. \tag{17}$$

Therefore, combining equation 16 and equation 17 and substituting in equation 15, as $n \to \infty$

$$\tilde{R}_{\text{odd}} \xrightarrow{\text{a.s.}} \frac{3^{p-1} \mathcal{L}_p(f^*) + 3^{p-1} \cdot 2^p \mathcal{L}_p(f^*)}{2}.$$

Exactly following a similar argument,

$$\tilde{R}_{\text{even}} \xrightarrow{\text{a.s.}} \frac{3^{p-1} \mathcal{L}_p(f^*) + 3^{p-1} \cdot 2^p \mathcal{L}_p(f^*)}{2}.$$

Therefore,

$$\tilde{R} \xrightarrow{\text{a.s.}} 3^{p-1} \mathcal{L}_p(f^*) + 3^{p-1} \cdot 2^p \mathcal{L}_p(f^*).$$

Using $\tilde{R} - \hat{R} \xrightarrow{\text{a.s.}} 0$ by Lemma B.2, as $n \to \infty$, we have $\hat{R} \xrightarrow{\text{a.s.}} 3^{p-1} \mathcal{L}_p(f^*) + 3^{p-1} \cdot 2^p \mathcal{L}_p(f^*)$. Finally, using the fact that $R \leq \hat{R}$, we obtain the following:

$$\lim_{n \to \infty} \mathbb{P}_S \left[ R \leq \left( 3^{p-1} \left( 2^p + 1 \right) + 1 \right) \mathcal{L}_p(f^*) \right] = 1. \tag{18}$$

Recall the definition of $R$ in equation 14 and $\mathcal{R}_p(\hat{g}_S)$ in equation 13. Having a probabilistic bound on $R$ gives us a bound on $\mathcal{R}_p(\hat{g}_S)$ when using Lemma B.1. In particular, we get

$$\lim_{n \to \infty} \mathbb{P}_S \left[ \mathcal{R}_p(\hat{g}_S) \leq \left( 3^{p-1} \left( 2^p + 1 \right) + 2 \right) \mathcal{L}_p(f^*) \right] = 1. \tag{19}$$

Finally, combining this with equation 12:

$$\lim_{n \to \infty} \mathbb{P}_S \left[ \mathcal{L}_p(\hat{g}_S) \leq C_p \, \mathcal{L}_p(f^*) \right] = 1,$$

where $C_p := 2^{p-1}[3^{p-1}((2^p + 1) + 2) + 1]$. $\qquad \square$

We now prove Lemmas B.1 and B.2 in order.

*Proof of Lemma B.1.* For any $x \in [0, x_1]$, we have $\hat{g}_S(x) = y_1$. Therefore,

$$|\hat{g}_S(x) - f^*(x)| = |y_1 - f^*(x)| = |f^*(x_1) + \epsilon_1 - f^*(x)| \leq G|x_1 - x| + |\epsilon_1| \leq G\ell_0 + |\epsilon_1|.$$

The implies that

$$R_0 = \int_0^{x_1} |\hat{g}_S(x) - f^*(x)|^p \, dx \leq (G\ell_0 + |\epsilon_1|)^p \ell_0 \leq 2^{p-1} G^p \ell_0^{p+1} + 2^{p-1} |\epsilon_1|^p \ell_0.$$

Similarly, for $x \in [x_n, 1]$

$$|\hat{g}_S(x) - f^*(x)| = |y_n - f^*(x)| = |f^*(x_n) + \epsilon_n - f^*(x)| \leq G|x_n - x| + |\epsilon_n| \leq G\ell_n + |\epsilon_n|.$$

Therefore

$$R_n = \int_{x_n}^1 |\hat{g}_S(x) - f^*(x)|^p \, dx \leq (G\ell_n + |\epsilon_n|)^p \ell_n \leq 2^{p-1} G^p \ell_n^{p+1} + 2^{p-1} |\epsilon_n|^p \ell_n.$$

Combining the two we get

$$\mathbb{E}_S[R_0 + R_n] \leq 2^{p-1}G^p \cdot (\mathbb{E}[\ell_0^{p+1}] + \mathbb{E}[\ell_n^{p+1}]) + 2^{p-1}(\mathbb{E}[|\epsilon_1|^p \ell_0] + \mathbb{E}[|\epsilon_n|^p \ell_n])$$

$$= 2^p G^p \mathbb{E}[\ell_0^{p+1}] + 2^p \mathcal{L}_p(f^*) \mathbb{E}[\ell_0] = 2^p G^p \mathbb{E}[\ell_0^{p+1}] + \frac{2^p \mathcal{L}_p(f^*)}{n+1}$$

$$\leq 2^p G^p \cdot \mathbb{E}\left[\frac{X_0^2}{(X_1 + \cdots + X_n)^2}\right] + \frac{2^p \mathcal{L}_p(f^*)}{n+1}$$

$$= 2^p G^p \cdot \mathbb{E}[\mathrm{Exp}(1)^2] \cdot \mathbb{E}\left[\frac{1}{\Gamma(n,1)^2}\right] + o_n(1)$$

$$= 2^p G^p \cdot 2 \cdot \int_0^\infty \frac{1}{z^2} \cdot \frac{z^{n-1} \cdot e^{-z}}{\Gamma(n)} \, dz + o_n(1) = \frac{2^{p+1}G^p}{\Gamma(n)} \cdot \int_0^\infty z^{n-3} \cdot e^{-z} \, dz + o_n(1)$$

$$= \frac{2^{p+1}G^p \Gamma(n-2)}{\Gamma(n)} + o_n(1) = \frac{2^{p+1}G^p}{(n-1)(n-2)} + o_n(1) = o_n(1).$$

Therefore, applying Markov's inequality yields that for any $\gamma > 0$,

$$\mathbb{P}_S[R_0 + R_n > \gamma] \leq \frac{\mathbb{E}[R_0 + R_n]}{\gamma} \leq o_n(1),$$

and the lemma follows. $\qquad\square$

*Proof of Lemma B.2.* By equation 9, we have $(\ell_0, \ldots, \ell_n) \sim \left(\frac{X_0}{X}, \ldots, \frac{X_n}{X}\right)$, where $X_0, \ldots, X_n \overset{\text{i.i.d.}}{\sim} \mathrm{Exp}(1)$ and $X := \sum_{i=0}^n X_i$. Also, we have $\tilde{\ell}_i = \frac{X_i}{n+1}$ for all $i$.

For every $p \geq 1$ we have

$$\sum_{i=1}^{n-1} \left(\tilde{\ell}_i^{p+1} - \ell_i^{p+1}\right) = \sum_{i=1}^{n-1} \left(\frac{X_i^{p+1}}{(n+1)^{p+1}} - \frac{X_i^{p+1}}{X^{p+1}}\right)$$

$$= \sum_{i=1}^{n-1} \frac{X_i^{p+1}}{(n+1)^{p+1}} \left(1 - \frac{(n+1)^{p+1}}{X^{p+1}}\right)$$

$$= \frac{n-1}{(n+1)^{p+1}} \cdot \frac{\sum_{i=1}^{n-1} X_i^{p+1}}{n-1} \left[1 - \left(\frac{n+1}{X}\right)^{p+1}\right],$$

and by the strong law of large numbers we have $\frac{\sum_{i=1}^{n-1} X_i^{p+1}}{n-1} \xrightarrow{\text{a.s.}} \mathbb{E}\, X_i^{p+1} < \infty$ and $\left(\frac{n+1}{X}\right)^{p+1} \xrightarrow{\text{a.s.}}$ 1, and thus

$$\sum_{i=1}^{n-1} \left(\tilde{\ell}_i^{p+1} - \ell_i^{p+1}\right) \xrightarrow{\text{a.s.}} 0.$$

Moreover, we have

$$\sum_{i=1}^{n-1} \left(\tilde{\ell}_i |\epsilon_i|^p - \ell_i |\epsilon_i|^p\right) = \sum_{i=1}^{n-1} \left(\frac{X_i |\epsilon_i|^p}{n+1} - \frac{X_i |\epsilon_i|^p}{X}\right)$$

$$= \sum_{i=1}^{n-1} \frac{X_i |\epsilon_i|^p}{n+1} \left(1 - \frac{n+1}{X}\right)$$

$$= \frac{n-1}{n+1} \cdot \frac{\sum_{i=1}^{n-1} X_i |\epsilon_i|^p}{n-1} \left(1 - \frac{n+1}{X}\right),$$

and by the strong law of large numbers we have $\frac{\sum_{i=1}^{n-1} X_i |\epsilon_i|^p}{n-1} \xrightarrow{\text{a.s.}} \mathbb{E}\, X_i |\epsilon_i|^p < \infty$ and $\frac{n+1}{X} \xrightarrow{\text{a.s.}} 1$, and thus

$$\sum_{i=1}^{n-1} \left(\tilde{\ell}_i |\epsilon_i|^p - \ell_i |\epsilon_i|^p\right) \xrightarrow{\text{a.s.}} 0.$$

Similarly, we also have

$$\sum_{i=1}^{n-1} \left( \tilde{\ell}_i |\epsilon_{i+1}|^p - \ell_i |\epsilon_{i+1}|^p \right) \xrightarrow{\text{a.s.}} 0 \ .$$

Overall, we get

$$\tilde{R} - \hat{R}$$

$$= \sum_{i=1}^{n-1} \Big( 3^{p-1}(2G)^p \tilde{\ell}_i^{p+1} + 3^{p-1} \tilde{\ell}_i |\epsilon_{i+1}|^p + 3^{p-1} \cdot 2^p \cdot \tilde{\ell}_i |\epsilon_i|^p$$

$$-3^{p-1}(2G)^p \ell_i^{p+1} - 3^{p-1} \ell_i |\epsilon_{i+1}|^p - 3^{p-1} \cdot 2^p \cdot \ell_i |\epsilon_i|^p \Big)$$

$$= 3^{p-1}(2G)^p \sum_{i=1}^{n-1} \left( \tilde{\ell}_i^{p+1} - \ell_i^{p+1} \right) + 3^{p-1} \sum_{i=1}^{n-1} \left( \tilde{\ell}_i |\epsilon_{i+1}|^p - \ell_i |\epsilon_{i+1}|^p \right) + 3^{p-1} \cdot 2^p \sum_{i=1}^{n-1} \left( \tilde{\ell}_i |\epsilon_i|^p - \ell_i |\epsilon_i|^p \right)$$

$$\xrightarrow{\text{a.s.}} 0 \ .$$

$\square$

## C  OMITTED PROOF FROM SECTION 4.1

Recall the definition of the affine functions $g_1, \ldots, g_{n-1}$ in Section 4.1. Also, recall the slope values $\delta_0, \ldots, \delta_n$ and the sign of the discrete curvature $\mathrm{curv}(x_i)$ at any point $x_i$. As mentioned, Boursier & Flammarion (2023) showed Lemma 2.1, which says that there is a unique minimizer of equation 2, which is piecewise linear and has at most one kink in the range $[x_i, x_{i+1})$. Due to this, we have only one degree of freedom between any two points; the solution can be completely characterized by variables $s^* = \{s_i^*\}_{i=1}^n$ where $s_i^*$ is the slope of the line incoming to point $(x_i, y_i)$. Formally, $s_i^* = \lim_{\epsilon \to 0^+} D\hat{f}(x_i - \epsilon)$.

Boursier & Flammarion (2023) (Lemma 3) proved the following lemma which upper and lower bounds the value of $s_i^*$ in terms of $\delta_{i-1}$ and $\delta_i$.

**Lemma C.1** (Boursier & Flammarion (2023)). *For any $i \in [n]$, $s_i^* \in [\min(\delta_{i-1}, \delta_i), \max(\delta_{i-1}, \delta_i)]$ where $\delta_0 = \delta_1$ and $\delta_n = \delta_{n-1}$.*

Having the above lemma at our disposal, we will now show Lemma 4.2, which characterizes the worst-case behavior of formation spikes.

*Proof of Lemma 4.2.* It is easy to see that the function $\hat{f}_S$ cannot have any kink on $(-\infty, x_1)$. This just follows from Lemma 2.1. Moreover, the slope of this straight line incoming to $(x_1, y_1)$ is given by $s_1^* \in [\min\{\delta_0, \delta_1\}, \max\{\delta_0, \delta_1\}] = \delta_1$ by Lemma C.1. Therefore, $\hat{f}_S(x)$ in the interval $(-\infty, x_1)$ must be $g_1(x)$. Now, again by Lemma 2.1 we can have at most one kink between $[x_1, x_2]$. Since $g_1(x)$ is a unique line joining the points $(x_1, y_1)$ and $(x_2, y_2)$, having one kink and changing the line at some point $x \in [x_1, x_2]$ will not pass through the point $(x_2, y_2)$. But since $\hat{f}_S(x_2) = y_2$, we must have that $\hat{f}_S(x) = g_1(x)$ in the entire $(-\infty, x_2)$.

Similarly, by Lemma 2.1, we don't have any kink from $[x_n, \infty)$. Moreover, the slope incoming to the point $(x_n, y_n)$ is $s_n^* = \delta_{n-1}$ since $s_n^* \in [\min\{\delta_{n-1}, \delta_n\}, \max\{\delta_{n-1}, \delta_n\}]$ and $\delta_{n-1} = \delta_n$ by Lemma C.1. Thus, it must be that $\hat{f}_S(x) = g_{n-1}(x)$ in $[x_n, \infty)$. Moreover, $\hat{f}_S(x)$ has at most one kink in $[x_{n-1}, x_n)$. This kink cannot belong to $(x_{n-1}, x_n)$ since $g_{n-1}(x)$ is the unique line joining the points $(x_{n-1}, y_{n-1})$ and $(x_n, y_n)$. Combining, we can say that $\hat{f}_S(x) = g_{n-1}(x)$ in $[x_{n-1}, \infty)$.

We now consider $[x_i, x_{i+1})$ for any $i \in \{2, \ldots, n-2\}$. We prove the lemma under three different cases.

1. **Case 1:** $\mathrm{curv}(x_i) = \mathrm{curv}(x_{i+1}) = -1$
   First of all, since $\delta_{i-1} > \delta_i$ and $g_{i-1}(x_i) = g_i(x_i) = y_i$, we have $g_{i-1}(x) \geq g_i(x)$ in $x \in [x_i, x_{i+1})$. Similarly, since $\delta_{i+1} < \delta_i$ even $g_{i+1}(x) \geq g_i(x)$ for $x \in [x_i, x_{i+1})$.

Using the same argument, since $s_i^* \in [\delta_i, \delta_{i-1}]$ and $s_{i+1}^* \in [\delta_{i+1}, \delta_i]$ by Lemma C.1, we say that $\hat{f}_S$ lies above $g_i(x)$ in $[x_i, x_{i+1})$. Therefore, it only remains to show that $\hat{f}_S(x) \le \min\{g_{i-1}(x), g_{i+1}(x)\}$.

Let us assume that it is not true. Let $x^* \in [x_i, x_{i+1})$ be the intersection point of $g_{i-1}(x)$ and $g_{i+1}(x)$. If $\hat{f}_S(x) \ge \min\{g_{i-1}(x), g_{i+1}(x)\}$ for some $x \in [x_i, x_{i+1})$, then it is easy to observe that it must be true at the location of the kink which is $x'$, i.e. $\hat{f}_S(x') \ge \min\{g_{i-1}(x'), g_{i+1}(x')\}$.

Now we have two possibilities; the first one is $x' < x^*$, in which case

$$s_i^* = \frac{\hat{f}_S(x') - y_i}{x' - x_i} > \frac{g_{i-1}(x') - y_i}{x' - x_i} = \delta_{i-1},$$

contradicting Lemma C.1. If $x' > x^*$ then

$$s_{i+1}^* = \frac{y_{i+1} - \hat{f}_S(x')}{x_{i+1} - x'} < \frac{y_{i+1} - g_{i+1}(x')}{x_{i+1} - x'} = \delta_{i+1},$$

again contradicting Lemma C.1. In either case, we get a contradiction and therefore, we must have

$$g_i(x) \le \hat{f}_S(x) \le \min\{g_{i-1}(x), g_{i+1}(x)\}.$$

2. **Case 2:** $\mathrm{curv}(x_i) = \mathrm{curv}(x_{i+1}) = +1$
Using a similar strategy, one can also show the desired result in this case. More formally, flip the label signs and apply exactly the same argument but on $-\hat{f}_S, -g_i, -g_{i-1}, -g_{i+1}$ instead. Then using the above argument, we achieve

$$-g_i(x) \le -\hat{f}_S(x) \le \min\{-g_{i-1}(x), -g_{i+1}(x)\}.$$

This implies

$$g_i(x) \ge \hat{f}_S(x) \ge \max\{g_{i-1}(x), g_{i+1}(x)\}.$$

3. **Case 3:** Otherwise, we may have several situations. We consider them one by one. If $\mathrm{curv}(x_i) = 0$. Then $\delta_{i-1} = \delta_i$. Then by Lemma C.1 we must have that $s_i^* = \delta_i$. Also, we have either one or no kink in $[x_i, x_{i+1})$ by Lemma 2.1. Since $g_i(x)$ is the unique line joining the points $(x_i, y_i)$ and $(x_{i+1}, y_{i+1})$, we cannot have any kink and $\hat{f}_S = g_i(x)$. Similarly, if $\mathrm{curv}(x_{i+1}) = 0$ then $\delta_i = \delta_{i+1}$ and $s_{i+1}^* = \delta_i$, which gives us that $\hat{f}_S(x) = g_i(x)$ using the uniqueness.

The only remaining possibilities are $\mathrm{curv}(x_i) = +1$ but $\mathrm{curv}(x_{i+1}) = -1$ or $\mathrm{curv}(x_i) = -1$ but $\mathrm{curv}(x_{i+1}) = +1$. W.l.o.g., we consider the former situation. Then $\delta_{i-1} < \delta_i$ and $\delta_i > \delta_{i+1}$. Also, by Lemma 2.1 there is at most one kink in $[x_i, x_{i+1})$. Therefore, if we show that the kink is at $x_i$ only, it is sufficient. Let us assume that it is not the case, namely, the kink is at $x' \in (x_i, x_{i+1})$. If $\hat{f}_S(x') > g_i(x)$, then the slope of the line incoming at $x_i$:

$$s_i^* = \frac{\hat{f}_S(x') - y_i}{x' - x_i} > \frac{g_i(x') - y_i}{x' - x_i} = \delta_i,$$

contradicting Lemma C.1. On the other hand, if $\hat{f}_S(x') < g_i(x)$, then the slope of the line incoming at $x_{i+1}$:

$$s_{i+1}^* = \frac{y_{i+1} - \hat{f}_S(x')}{x_{i+1} - x'} > \frac{y_{i+1} - g_i(x')}{x_{i+1} - x'} = \delta_i,$$

contradicting Lemma C.1.

Therefore, the lemma is true in all the cases.

$\square$

As a corollary, we get the following lemma.

**Lemma C.2.** *Fix any* $i \in \{2, \ldots, n-2\}$, *and consider* $x \in [x_i, x_{i+1})$:

$$|\hat{f}_S(x) - f^*(x)| \leq \max\left\{|g_i(x) - f^*(x)|, \min\{|g_{i+1}(x) - f^*(x)|, |g_{i-1}(x) - f^*(x)|\}\right\},$$

*and for* $x \in [0, x_2)$

$$|\hat{f}_S(x) - f^*(x)| = |g_1(x) - f^*(x)|,$$

*and for* $x \in [x_{n-1}, 1]$

$$|\hat{f}_S(x) - f^*(x)| = |g_{n-1}(x) - f^*(x)|.$$

*Proof.* The above lemma is clearly true for $x \in [0, x_2)$, since in that range, $\hat{f}_S(x) = g_1(x)$ by Lemma 4.2. Similarly, for $x \in [x_{n-1}, 1]$ we have $\hat{f}_S(x) = g_{n-1}(x)$ by Lemma 4.2. This implies that $|\hat{f}_S(x) - f^*(x)| = |g_{n-1}(x) - f^*(x)|$.

Also, by applying Lemma 4.2, for any $i \in \{2, \ldots, n-2\}$ and for any $x \in [x_i, x_{i+1})$, one can say that unless $\mathsf{curv}(x_i) = \mathsf{curv}(x_{i+1}) = -1$ or $\mathsf{curv}(x_i) = \mathsf{curv}(x_{i+1}) = +1$, we have

$$\hat{f}_S(x) = |g_i(x) - f^*(x)|,$$

and the lemma holds. If $\mathsf{curv}(x_i) = \mathsf{curv}(x_{i+1}) = -1$. Then by Lemma 4.2, we have $g_i(x) \leq \hat{f}_S(x) \leq \min\{g_{i-1}(x), g_{i+1}(x)\}$. Let $x^* \in [x_i, x_{i+1})$, where $g_{i-1}(x)$ and $g_{i+1}(x)$ meet.

For $x \in [x_i, x^*]$: $g_i(x) \leq \hat{f}_S(x) \leq g_{i-1}(x) \leq g_{i+1}(x)$. Therefore,

$$\underbrace{g_i(x) - f^*(x)}_{:=(1)} \leq \hat{f}_S(x) - f^*(x) \leq \underbrace{g_{i-1}(x) - f^*(x)}_{:=(2)} \leq \underbrace{g_{i+1}(x) - f^*(x)}_{:=(3)}.$$

If $(1)$ is non-negative, then the claim is clearly true. Because even $(2)$ and $(3)$ are positive. If $(1)$ is negative then the only way the $|\hat{f}_S(x) - f^*(x)|$ can be greater than $|g_i(x) - f^*(x)|$ is when $g_{i-1}(x) - f^*(x)$ is positive. And thus $g_{i+1}(x) - f^*(x)$ is also positive. Therefore we have

$$-|g_i(x) - f^*(x)| \leq \hat{f}_S(x) - f^*(x) \leq |g_{i-1}(x) - f^*(x)| \leq |g_{i+1}(x) - f^*(x)|,$$

which immediately implies the desired result. Similarly, for $x \in (x^*, x_{i+1})$: $g_i(x) \leq \hat{f}_S(x) \leq g_{i+1}(x) \leq g_{i-1}(x)$. Therefore,

$$\underbrace{g_i(x) - f^*(x)}_{:=(1)} \leq \hat{f}_S(x) - f^*(x) \leq \underbrace{g_{i+1}(x) - f^*(x)}_{:=(2)} \leq \underbrace{g_{i-1}(x) - f^*(x)}_{:=(3)}.$$

Again if $(1)$ is non-negative, then the claim is clearly true. If $(1)$ is negative then the only way the $|\hat{f}_S(x) - f^*(x)|$ can be greater than $|g_i(x) - f^*(x)|$ is when $g_{i+1}(x) - f^*(x)$ is positive. And thus $g_{i-1}(x) - f^*(x)$ is also positive. Therefore we have

$$-|g_i(x) - f^*(x)| \leq \hat{f}_S(x) - f^*(x) \leq |g_{i+1}(x) - f^*(x)| \leq |g_{i-1}(x) - f^*(x)|,$$

and the lemma follows. The proof is exactly symmetric for when $\mathsf{curv}(x_i) = \mathsf{curv}(x_i) = +1$; the only difference is that we apply the same argument on $-g_i, -g_{i-1}, -g_{i+1}$ and $-\hat{f}_S$ and add the function $f^*(x)$ instead of subtracting.

$\square$

We now recall Definition 4.3 of special points. Boursier & Flammarion (2023) proved that if two special points occur within two points. Then we get a spike. Our Lemma 4.4 is a mild generalization of the lemma.

*Proof of Lemma 4.4.* (Boursier & Flammarion, 2023, Lemma 8) already showed that if $n_{k+1} = n_k + 2$ then $\hat{f}_S$ has exactly one kink in $(x_{n_k-1}, x_{n_k+1})$. Consider four consecutive points $(x_{n_k-1}, y_{n_k-1})$, $(x_{n_k}, y_{n_k})$, $(x_{n_k+1}, y_{n_k+1})$, and $(x_{n_k+2}, x_{n_k+2})$. The first three are non-collinear and even the last three are non-colinear since $\mathsf{curv}(x_{n_k}) = \mathsf{curv}(x_{n_k+1}) = -1$. Therefore, the only way to interpolate them with 2 linear pieces is to extend $g_{n_k-1}$ and $g_{n_k+1}$ until they intersect. This immediately implies that $\hat{f}_S(x) = \min\{g_{n_k-1}, g_{n_k+1}\}$.

$\square$

## D  PROOF OF THEOREM 2

*Proof of Theorem 2.* Let $f^*$ be $G$-Lipschitz. We sample $S \sim \mathcal{D}^n$ and number points $(x_i, y_i)$ from left to right based on the $x$-coordinate. Again, we denote $x_0 := 0$ and $x_{n+1} := 1$ for notational convenience (they are not used in determining $\hat{f}_S$). The domain $[0, 1]$ is divided into $n + 1$ intervals of length $\ell_0, \ell_1, \dots, \ell_n$, where $\ell_i = x_{i+1} - x_i$. We want to analyze the population $L_p$ loss of the min-norm interpolator $\hat{f}_S$ as defined in equation 2 for $p \in [1, 2)$. Exactly following the step for the derivation of equation 48, we get

$$\mathcal{L}_p(\hat{f}_S) \le 2^{p-1}(\mathcal{R}_p(\hat{f}_S) + \mathcal{L}_p(f^*)), \tag{20}$$

where $\mathcal{R}_p(\hat{f}_S)$ is defined and simplified as the following.

$$
\begin{aligned}
\mathcal{R}_p(\hat{f}_S) &= \int_{x_0=0}^{x_{n+1}=1} |\hat{f}_S(x) - f^*(x)|^p \, dx \\
&= \sum_{i=0}^{n} \int_{x_i}^{x_{i+1}} |\hat{f}_S(x) - f^*(x)|^p \, dx \\
&\le \int_{0}^{x_2} |g_1(x) - f^*(x)|^p \, dx + \int_{x_{n-1}}^{1} |g_{n-1}(x) - f^*(x)|^p \, dx \\
&\quad + \sum_{i=2}^{n-2} \int_{x_i}^{x_{i+1}} \max\{|g_i(x) - f^*(x)|^p, \min\{|g_{i-1}(x) - f^*(x)|^p, |g_{i+1}(x) - f^*(x)|^p\}\} \, dx \\
&\qquad\qquad\qquad\qquad\qquad\qquad\qquad\qquad\qquad\qquad\qquad \text{(by Lemma C.2)} \\
&\le \int_{0}^{x_2} |g_1(x) - f^*(x)|^p \, dx + \int_{x_{n-1}}^{1} |g_{n-1}(x) - f^*(x)|^p \, dx \\
&\quad + \sum_{i=2}^{n-2} \int_{x_i}^{x_{i+1}} |g_i(x) - f^*(x)|^p + \min\{|g_{i-1}(x) - f^*(x)|^p, |g_{i+1}(x) - f^*(x)|^p\} \, dx \\
&\le \underbrace{\int_{0}^{x_1} |g_1(x) - f^*(x)|^p \, dx}_{:=R_0} + \underbrace{\int_{x_n}^{1} |g_{n-1}(x) - f^*(x)|^p \, dx}_{:=R_n} + \sum_{i=1}^{n-1} \int_{x_i}^{x_{i+1}} |g_i(x) - f^*(x)|^p \, dx \\
&\quad + \underbrace{\sum_{i=2}^{n-2} \int_{x_i}^{x_{i+1}} \min\{|g_{i-1}(x) - f^*(x)|^p, |g_{i+1}(x) - f^*(x)|^p\} \, dx}_{:=R} \\
&\le R_0 + R_n + \mathcal{R}_p(\hat{g}_S) + R. \tag{21}
\end{aligned}
$$

The following lemma, which we prove after the theorem, establishes that $R_0 + R_n$ is vanishing with high probability.

**Lemma D.1.** *For any $\gamma > 0$, we have $\lim_{n \to \infty} \mathbb{P}_S[R_0 + R_n \le \gamma] = 1$.*

Moreover, we have already bounded $\mathcal{R}_p(\hat{g}_S)$ in equation 19; in fact for any $p \ge 1$. When $p \in [1, 2)$, it reduces to:

$$\lim_{n \to \infty} \mathbb{P}_S [\mathcal{R}_p(\hat{g}_S) \le 17 \, \mathcal{L}_p(f^*)] = 1. \tag{22}$$

We now focus on bounding $R$. Intuitively, $R$ is the risk term caused due to spike formation. This is only bounded for $p \in [1, 2)$ and grows as $p$ approaches 2. For $i \in \{2, \dots, n-2\}$, define:

$$R_i := \int_{x_i}^{x_{i+1}} \min\{|g_{i-1}(x) - f^*(x)|^p, |g_{i+1}(x) - f^*(x)|^p\} \, dx.$$

Then $R = \sum_{i=2}^{n-2} R_i$. Moreover, each $R_i$ can be simplified as the following. For any $i \in \{2, \ldots, n-2\}$ and any $x \in [x_i, x_{i+1}]$,

$$
\begin{aligned}
|g_{i-1}(x) - f^*(x)| &= \left| y_i + \frac{(y_i - y_{i-1})}{x_i - x_{i-1}}(x - x_i) - f^*(x) \right| \\
&= \left| f^*(x_i) + \epsilon_i + \left( \frac{f^*(x_i) + \epsilon_i - f^*(x_{i-1}) - \epsilon_{i-1}}{x_i - x_{i-1}} \right)(x - x_i) - f^*(x) \right| \\
&\leq |f^*(x_i) - f^*(x)| + |\epsilon_i| + \frac{G(x_i - x_{i-1}) + |\epsilon_i - \epsilon_{i-1}|}{x_i - x_{i-1}} |(x - x_i)| \\
&\leq G(x - x_i) + |\epsilon_i| + \left( \frac{G(x_i - x_{i-1}) + |\epsilon_i| + |\epsilon_{i-1}|}{x_i - x_{i-1}} \right)(x - x_i) \\
&\leq G \cdot \ell_i + |\epsilon_i| + \left( \frac{G \cdot \ell_{i-1} + |\epsilon_i| + |\epsilon_{i-1}|}{\ell_{i-1}} \right) \cdot \ell_i \\
&= 2G \cdot \ell_i + |\epsilon_i| + (|\epsilon_i| + |\epsilon_{i-1}|) \frac{\ell_i}{\ell_{i-1}}
\end{aligned}
\tag{23}
$$

$$
\begin{aligned}
|g_{i+1}(x) - f^*(x)| &= \left| y_{i+1} + \frac{(y_{i+2} - y_{i+1})}{x_{i+2} - x_{i+1}}(x - x_{i+1}) - f^*(x) \right| \\
&= \left| f^*(x_{i+1}) + \epsilon_{i+1} + \left( \frac{f^*(x_{i+2}) + \epsilon_{i+2} - f^*(x_{i+1}) - \epsilon_{i+1}}{x_{i+2} - x_{i+1}} \right)(x - x_{i+1}) - f^*(x) \right| \\
&\leq |f^*(x_{i+1}) - f^*(x)| + |\epsilon_{i+1}| + \frac{G(x_{i+2} - x_{i+1}) + |\epsilon_{i+2} - \epsilon_{i+1}|}{x_{i+2} - x_{i+1}} |x - x_{i+1}| \\
&\leq G|x - x_{i+1}| + |\epsilon_{i+1}| + \left( \frac{G(x_{i+2} - x_{i+1}) + |\epsilon_{i+2}| + |\epsilon_{i+1}|}{x_{i+2} - x_{i+1}} \right) |x - x_{i+1}| \\
&\leq G \cdot \ell_i + |\epsilon_{i+1}| + \left( \frac{G \cdot \ell_{i+1} + |\epsilon_{i+2}| + |\epsilon_{i+1}|}{\ell_{i+1}} \right) \cdot \ell_i \\
&= 2G \cdot \ell_i + |\epsilon_{i+1}| + (|\epsilon_{i+1}| + |\epsilon_{i+2}|) \frac{\ell_i}{\ell_{i+1}}
\end{aligned}
\tag{24}
$$

$$
\begin{aligned}
R_i &= \int_{x_i}^{x_{i+1}} \min\{|g_{i-1}(x) - f^*(x)|^p, |g_{i+1}(x) - f^*(x)|^p\} \, dx \\
&\leq \int_{x_i}^{x_{i+1}} \min\left\{ 2G \cdot \ell_i + |\epsilon_i| + (|\epsilon_i| + |\epsilon_{i-1}|)\frac{\ell_i}{\ell_{i-1}}, 2G \cdot \ell_i + |\epsilon_{i+1}| + (|\epsilon_{i+1}| + |\epsilon_{i+2}|)\frac{\ell_i}{\ell_{i+1}} \right\}^p dx \\
&\qquad\qquad\qquad\qquad\qquad \text{(using equation 23 and equation 24)} \\
&\leq \left( 2G\ell_i + |\epsilon_i| + |\epsilon_{i+1}| + (|\epsilon_i| + |\epsilon_{i+1}| + |\epsilon_{i-1}| + |\epsilon_{i+2}|)\frac{\ell_i}{\max\{\ell_{i-1}, \ell_{i+1}\}} \right)^p \ell_i, \\
&\leq 4^{p-1} \left( (2G\ell_i)^p + |\epsilon_i|^p + |\epsilon_{i+1}|^p + \left( (2|\epsilon_i| + 2|\epsilon_{i+1}| + |\epsilon_{i-1}| + |\epsilon_{i+2}|)\frac{\ell_i}{\max\{\ell_{i-1}, \ell_{i+1}\}} \right)^p \right) \ell_i \\
&= 2^{3p-2} G^p \ell_i^{p+1} + 4^{p-1}|\epsilon_i|^p \ell_i + 4^{p-1}|\epsilon_{i+1}|^p \ell_i + 4^{p-1}(|\epsilon_i| + |\epsilon_{i+1}| + |\epsilon_{i-1}| + |\epsilon_{i+2}|)^p \frac{\ell_i^{p+1}}{\max\{\ell_{i-1}, \ell_{i+1}\}^p} \\
&:= \hat{R}_i
\end{aligned}
\tag{25}
$$

Then $\hat{R} := \sum_{i=2}^{n-2} \hat{R}_i$ is an upper bound on $R$. The random variables $\ell_i$ s are mildly dependent. To achieve independence, we denote $\tilde{\ell}_0, \tilde{\ell}_1, \ldots, \tilde{\ell}_n \overset{\text{i.i.d.}}{\sim} \text{Exp}(1)/(n+1)$; in particular, $\tilde{\ell}_i = X_i/n+1$.

We replace $\ell_i$ with $\tilde{\ell}_i$ and define random variables $\tilde{R}_i$.

$$\tilde{R}_i = 2^{3p-2}G^p\tilde{\ell}_i^{p+1} + 4^{p-1}(|\epsilon_i|^p + |\epsilon_{i+1}|^p)\tilde{\ell}_i + 4^{p-1}(|\epsilon_i| + |\epsilon_{i+1}| + |\epsilon_{i-1}| + |\epsilon_{i+2}|)^p \frac{\tilde{\ell}_i^{p+1}}{\max\{\tilde{\ell}_{i-1}, \tilde{\ell}_{i+1}\}^p}, \text{ and}$$

$$\tilde{R} = \sum_{i=2}^{n-2} \tilde{R}_i.$$

We claim that:

**Lemma D.2.** *As $n \to \infty$, we have $\hat{R} - \tilde{R} \xrightarrow{a.s.} 0$ .*

The proof is similar to Lemma B.2. Therefore, it suffices to give a bound on $\tilde{R}$. Still, any four consecutive $\tilde{R}_i$'s are dependent. Therefore, we re-express $\tilde{R}$ into four disjoint sums such that each individual of them is the sum of only i.i.d. random variables. We divide the indices into four sets $\mathcal{I}_j$ for $0 \le j \le 3$. Define $\mathcal{I}_j = \{i\%4 = j : 2 \le i \le (n-2)\}$ for $0 \le j \le 3$ and

$$\tilde{R}^{(j)} := \sum_{i \in \mathcal{I}_j} \tilde{R}_i. \quad \text{Then} \quad \tilde{R} = \sum_{j=0}^{3} \tilde{R}^{(j)}.$$

Then for any $0 \le j \le 3$, further simplifying

$$\tilde{R}^{(j)} = \sum_{i \in \mathcal{I}_j} 2^{3p-2}G^p\tilde{\ell}_i^{p+1} + 4^{p-1}(|\epsilon_i|^p + |\epsilon_{i+1}|^p)\tilde{\ell}_i + 4^{p-1}(|\epsilon_i| + |\epsilon_{i+1}| + |\epsilon_{i-1}| + |\epsilon_{i+2}|)^p \frac{\tilde{\ell}_i^{p+1}}{\max\{\tilde{\ell}_{i-1}, \tilde{\ell}_{i+1}\}^p}$$

$$= \underbrace{\frac{1}{(n+1)^{p+1}} \sum_{i \in \mathcal{I}_j} 2^{3p-2}G^p X_i^{p+1}}_{:=T_1} + \underbrace{\frac{1}{(n+1)} \sum_{i \in \mathcal{I}_j} 4^{p-1}(|\epsilon_i|^p + |\epsilon_{i+1}|^p)X_i}_{:=T_2}$$

$$+ \underbrace{\frac{1}{(n+1)} \sum_{i \in \mathcal{I}_j} 4^{p-1}(|\epsilon_i| + |\epsilon_{i+1}| + |\epsilon_{i-1}| + |\epsilon_{i+2}|)^p \frac{X_i^{p+1}}{\max\{X_{i-1}, X_{i+1}\}^p}}_{:=T_3} \qquad (26)$$

Now, each $T_1, T_2$ and $T_3$ is the average of i.i.d. random variables (up to scaling). Thus, as $n \to \infty$, by the strong LLN,

$$\frac{1}{|\mathcal{I}_j|} \sum_{i \in \mathcal{I}_j} 2^{3p-2}G^p X_i^{p+1} \xrightarrow{a.s.} 2^{3p-2}G^p \mathbb{E}[X_i^{p+1}] = 2^{3p-2}G^p\Gamma(p+2) < \infty.$$

Therefore, using the fact $\lim_{n \to \infty} \frac{|\mathcal{I}_j|}{(n+1)} = \frac{1}{4}$ and since we are considering that $p \ge 1$ we get that the first term of equation 26 converges to 0 almost surely, i.e. as $n \to \infty$

$$T_1 = \frac{1}{(n+1)^{p+1}} \sum_{i \in \mathcal{I}_j} 2^{3p-2}G^p X_i^{p+1} \xrightarrow{a.s.} 0. \qquad (27)$$

Similarly, since $\mathbb{E}[X_i] = 1$ and $\mathbb{E}[|\epsilon|^p] = \mathcal{L}_p(f^*) < \infty$ even $T_2$ (up to scaling) is the average i.i.d. random variables, with finite expectation. Using the strong law of large numbers, as $n \to \infty$ the second term of equation 26

$$T_2 = \frac{1}{(n+1)} \sum_{i \in \mathcal{I}_j} 4^{p-1}(|\epsilon_i|^p + |\epsilon_{i+1}|^p)X_i \xrightarrow{a.s.} \frac{1}{4} \cdot 4^{p-1} \mathbb{E}[(|\epsilon_i|^p + |\epsilon_{i+1}|^p)X_i] = \frac{4^{p-1}}{2}\mathcal{L}_p(f^*).$$

$$(28)$$

Finally, $T_3$ is also the average of i.i.d random variables (up to scaling). Before applying the strong LLN, we must verify if each summand has a finite expectation. Since $\mathbb{E}[|\epsilon|^p] = \mathcal{L}_p(f^*) < \infty$ and $\mathbb{E}[X_i^{p+1}] = \Gamma(p+2) < \infty$, it is easy to observe that it boils down to verifying if $\mathbb{E}[\frac{1}{\max\{X_{i-1}, X_{i+1}\}^p}]$ has a finite expectation. The following claim verifies this.

**Claim D.3.** *Let $A, B \overset{i.i.d.}{\sim} Exp(1)$, then $\mathbb{E}\left[\frac{1}{\max\{A,B\}^p}\right] \le \frac{2^p}{(2-p)}$.*

Therefore, applying the strong LLN in exactly the same way:

$$T_3 = \frac{1}{(n+1)} \sum_{i \in \mathcal{I}_j} 4^{p-1} (|\epsilon_i| + |\epsilon_{i+1}| + |\epsilon_{i-1}| + |\epsilon_{i+2}|)^p \frac{X_i^{p+1}}{\max\{X_{i-1}, X_{i+1}\}^p} \xrightarrow{\text{a.s.}}$$

$$\frac{1}{4} \cdot \mathbb{E} \left[ 4^{p-1} (|\epsilon_i| + |\epsilon_{i+1}| + |\epsilon_{i-1}| + |\epsilon_{i+2}|)^p \frac{X_i^{p+1}}{\max\{X_{i-1}, X_{i+1}\}^p} \right]$$

$$\leq \frac{4^{p-1} 4^{p-1} (\mathbb{E}[|\epsilon_i|^p + |\epsilon_{i+1}|^p + |\epsilon_{i-1}|^p + |\epsilon_{i+2}|^p])}{4} \Gamma(p+2) \cdot \mathbb{E} \left[ \frac{1}{\max\{X_{i-1}, X_{i+1}\}^p} \right]$$

$$\leq \frac{2^{4p-4} \mathcal{L}_p(f^*) \Gamma(p+2) 2^p}{(2-p)}, \tag{29}$$

where in the last inequality, we use Claim D.3. Using equation 29, equation 27 and equation 28, we get a high probability bound on $\tilde{R}^{(j)}$ (recall definition in equation 26). In particular, for $0 \leq j \leq 3$

$$\lim_{n \to \infty} \mathbb{P}_S \left[ \tilde{R}^{(j)} \leq \left( \frac{4^{p-1}}{2} + \frac{2^{4p-4} \cdot 2^p \cdot \Gamma(p+2)}{2-p} + 1 \right) \mathcal{L}_p(f^*) \right] = 1.$$

Using the fact that we are considering $p \in [1, 2)$, we get

$$\lim_{n \to \infty} \mathbb{P}_S \left[ \tilde{R}^{(j)} \leq \left( 2 + \frac{384}{2-p} + 1 \right) \mathcal{L}_p(f^*) \right] = 1,$$

$$\implies \lim_{n \to \infty} \mathbb{P}_S \left[ \tilde{R}^{(j)} \leq \frac{387 \mathcal{L}_p(f^*)}{2-p} \right] = 1.$$

Therefore, since $\tilde{R} = \sum_{j=0}^3 \tilde{R}^{(j)}$ we obtain

$$\lim_{n \to \infty} \mathbb{P}_S \left[ \tilde{R} \leq \frac{1548 \, \mathcal{L}_p(f^*)}{(2-p)} \right] = 1.$$

Using this along with Lemma D.2 and the fact that $R \leq \hat{R}$ gives us

$$\lim_{n \to \infty} \mathbb{P}_S \left[ R \leq \frac{1549 \, \mathcal{L}_p(f^*)}{(2-p)} \right] = 1.$$

Further substituting this in the bounds from equation 21 and using Lemma D.1 and equation 22

$$\lim_{n \to \infty} \mathbb{P}_S \left[ \mathcal{R}_p(\hat{f}_S) \leq \frac{1567 \mathcal{L}_p(f^*)}{(2-p)} \right] = 1.$$

Finally, using equation 20 with the above

$$\lim_{n \to \infty} \mathbb{P}_S \left[ \mathcal{L}_p(\hat{f}_S) \leq \frac{3136 \mathcal{L}_p(f^*)}{(2-p)} \right] = 1.$$

As such, by letting $C = 3136$, the theorem follows. $\qquad \square$

We now prove the lemmas and claims we used in the above proof.

*Proof of Lemma D.1.* For any $x \in [0, x_1]$, we have

$$
\begin{aligned}
|g_1(x) - f^*(x)| &= \left| y_1 + \frac{(y_2 - y_1)}{x_2 - x_1}(x - x_1) - f^*(x) \right| \\
&= \left| f^*(x_1) + \epsilon_1 + \left( \frac{f^*(x_2) + \epsilon_2 - f^*(x_1) - \epsilon_1}{x_2 - x_1} \right)(x - x_1) - f^*(x) \right| \\
&\leq |f^*(x_1) - f^*(x)| + |\epsilon_1| + \left| \frac{G(x_2 - x_1) + |\epsilon_2 - \epsilon_1|}{x_2 - x_1} \right| |x - x_1| \\
&\leq G|x - x_1| + |\epsilon_1| + \left( \frac{G(x_2 - x_1) + |\epsilon_2| + |\epsilon_1|}{x_2 - x_1} \right) |x - x_1| \\
&\leq G \cdot \ell_0 + |\epsilon_1| + \left( \frac{G \cdot \ell_1 + |\epsilon_2| + |\epsilon_1|}{\ell_1} \right) \cdot \ell_0 \\
&= 2G \cdot \ell_0 + |\epsilon_1| + (|\epsilon_2| + |\epsilon_1|) \frac{\ell_0}{\ell_1}.
\end{aligned}
$$

Therefore,

$$
\begin{aligned}
R_0 = \int_0^{x_1} |g_1(x) - f^*(x)|^p \, dx &\leq \ell_0 \left[ 2G \cdot \ell_0 + |\epsilon_1| + (|\epsilon_2| + |\epsilon_1|) \frac{\ell_0}{\ell_1} \right]^p \\
&\leq 4^{p-1} \cdot (2G)^p \cdot \ell_0^{p+1} + 4^{p-1} |\epsilon_1|^p \ell_0 + 4^{p-1} |\epsilon_2|^p \frac{\ell_0^{p+1}}{\ell_1^p} + 4^{p-1} \cdot |\epsilon_1|^p \cdot \frac{\ell_0^{p+1}}{\ell_1^p} \\
&= 4^{p-1} \cdot (2G)^p \cdot \frac{X_0^{p+1}}{(X_0 + \cdots + X_n)^{p+1}} + 4^{p-1} |\epsilon_1|^p \frac{X_0}{(X_0 + \cdots + X_n)} \\
&\quad + (4^{p-1} \cdot |\epsilon_2|^p + 4^{p-1} \cdot |\epsilon_1|^p) \cdot \frac{X_0^{p+1}}{X_1^p \cdot (X_0 + \cdots + X_n)}
\end{aligned}
\tag{30}
$$

We now provide probabilistic upper bounds on $X_0$, $|\epsilon_2|^p$ and $|\epsilon_1|^p$, and lower bounds on $X_1$ and $X_0 + \cdots + X_n$, which suffices to further upper bound $R_0$.

$$
\mathbb{P}[X_0 \leq 3 \ln n] = 1 - \mathbb{P}[X_0 > 3 \ln n] = 1 - \int_{3 \ln n}^\infty e^{-z} \, dz = 1 - \frac{1}{n^3}.
$$

Using Markov's inequality,

$$
\mathbb{P}[|\epsilon_1|^p \leq \mathcal{L}_p(f^*) \ln n] = 1 - \mathbb{P}[|\epsilon_1|^p > \mathcal{L}_p(f^*) \ln n] \geq 1 - \frac{\mathbb{E}[|\epsilon_1|^p]}{\mathcal{L}_p(f^*) \ln n} = 1 - \frac{1}{\ln n}.
$$

Similarly,

$$
\mathbb{P}[|\epsilon_2|^p \leq \mathcal{L}_p(f^*) \ln n] \geq 1 - \frac{1}{\ln n}.
$$

$$
\mathbb{P}[X_1 \geq \frac{1}{\ln n}] = \int_{\frac{1}{\ln n}}^\infty e^{-z} \, dz = e^{-\frac{1}{\ln n}} \geq 1 - \frac{1}{\ln n},
$$

where the last inequality follows from the Taylor expansion of $e^z$. Finally, as $n \to \infty$, by the strong Law of Large Numbers, $\frac{1}{n} \sum_{i=0}^n X_i \xrightarrow{\text{a.s.}} 1$. This implies

$$
\mathbb{P}\left[ \sum_{i=0}^n X_i \geq \frac{n}{2} \right] = 1 - o_n(1).
$$

Doing a union bound over these events, and substituting these probabilistic bounds in equation 30, we get that with probability $1 - o_n(1)$,

$$
R_0 \leq \frac{4^{p-1} \cdot (2G)^p (3 \ln n)^{p+1}}{(n/2)^{p+1}} + \frac{4^{p-1} \mathcal{L}_p(f^*) \ln n \cdot 3 \ln n}{(n/2)} + \frac{2 \cdot 4^{p-1} \mathcal{L}_p(f^*) \ln n \cdot (3 \ln n)^{p+1}}{(1/\ln^p n) \cdot (n/2)} = o_n(1).
$$

Following the exact same steps, one can say that with probability $1 - o_n(1)$, we have $R_n = o_n(1)$. Therefore, doing the union bound and combining both, we get that with probability $1 - o_n(1)$ we have $R_0 + R_n = o_n(1)$. This implies, for any $\gamma > 0$,

$$
\lim_{n \to \infty} \mathbb{P}_S [R_0 + R_n \leq \gamma] = 1.
$$

$\square$

*Proof of Lemma D.2.* By equation 9, we have $(\ell_0, \ldots, \ell_n) \sim \left(\frac{X_0}{X}, \ldots, \frac{X_n}{X}\right)$, where $X_0, \ldots, X_n \overset{\text{i.i.d.}}{\sim} \text{Exp}(1)$ and $X := \sum_{i=0}^n X_i$. Also, we have $\tilde{\ell}_i = \frac{X_i}{n+1}$ for all $i$.

Therefore,

$$\sum_{i=2}^{n-2} 4^{p-1}(|\epsilon_i| + |\epsilon_{i+1}| + |\epsilon_{i-1}| + |\epsilon_{i+2}|)^p \left(\frac{\ell_i^{p+1}}{\max\{\ell_{i-1}, \ell_{i+1}\}^p} - \frac{\tilde{\ell}_i^{p+1}}{\max\{\tilde{\ell}_{i-1}, \tilde{\ell}_{i+1}\}^p}\right)$$

$$= \sum_{i=2}^{n-2} 4^{p-1}(|\epsilon_i| + |\epsilon_{i+1}| + |\epsilon_{i-1}| + |\epsilon_{i+2}|)^p \cdot \frac{X_i^{p+1}}{\max\{X_{i-1}, X_{i+1}\}^p} \left(\frac{1}{X} - \frac{1}{n+1}\right)$$

$$= \underbrace{\frac{1}{(n+1)} \sum_{i=2}^{n-2} 4^{p-1}(|\epsilon_i| + |\epsilon_{i+1}| + |\epsilon_{i-1}| + |\epsilon_{i+2}|)^p \cdot \frac{X_i^{p+1}}{\max\{X_{i-1}, X_{i+1}\}^p} \left(\frac{n+1}{X} - 1\right)}_{:=T}$$

By the strong law of large numbers, as $n \to \infty$, $\frac{n+1}{X} \overset{\text{a.s.}}{\longrightarrow} 1$. Thus, $\left(\frac{n+1}{X} - 1\right) \overset{\text{a.s.}}{\longrightarrow} 0$. Also, $T$ converges almost surely to a finite quantity as $n \to \infty$. To see this, we split $T$ into four disjoint sums, one each over indices in $\mathcal{I}_j = \{i : i\%4 = j\}$ for $0 \le j \le 3$. And each of them converges almost surely to a finite quantity by equation 29, and hence also the overall sum $T$. Therefore, as $n \to \infty$

$$\sum_{i=2}^{n-2} 4^{p-1}(|\epsilon_i| + |\epsilon_{i+1}| + |\epsilon_{i-1}| + |\epsilon_{i+2}|)^p \left(\frac{\ell_i^{p+1}}{\max\{\ell_{i-1}, \ell_{i+1}\}^p} - \frac{\tilde{\ell}_i^{p+1}}{\max\{\tilde{\ell}_{i-1}, \tilde{\ell}_{i+1}\}^p}\right) \overset{\text{a.s.}}{\longrightarrow} 0 . \quad (31)$$

Next, we have

$$\sum_{i=2}^{n-2} 2^{3p-2} G^p \left(\ell_i^{p+1} - \tilde{\ell}_i^{p+1}\right) = \sum_{i=2}^{n-2} 2^{3p-2} G^p X_i^{p+1} \left(\frac{1}{X^{p+1}} - \frac{1}{(n+1)^{p+1}}\right)$$

$$= 2^{3p-2} G^p \cdot \frac{\sum_{i=2}^{n-2} X_i^{p+1}}{n-3} \cdot \frac{n-3}{(n+1)^{p+1}} \cdot \left(\frac{(n+1)^{p+1}}{X^{p+1}} - 1\right) ,$$

and by the strong law of large numbers we have $\frac{\sum_{i=2}^{n-2} X_i^{p+1}}{n-3} \overset{\text{a.s.}}{\longrightarrow} \mathbb{E} X_i^{p+1} < \infty$ and $\left(\frac{n+1}{X}\right)^{p+1} \overset{\text{a.s.}}{\longrightarrow} 1$, and thus

$$\sum_{i=2}^{n-2} 2^{3p-2} G^p \left(\ell_i^{p+1} - \tilde{\ell}_i^{p+1}\right) \overset{\text{a.s.}}{\longrightarrow} 0 . \quad (32)$$

Moreover,

$$\sum_{i=2}^{n-2} 4^{p-1} |\epsilon_i|^p \left(\ell_i - \tilde{\ell}_i\right) = 4^{p-1} \sum_{i=2}^{n-2} \left(\frac{|\epsilon_i|^p X_i}{X} - \frac{|\epsilon_i|^p X_i}{n+1}\right)$$

$$= 4^{p-1} \cdot \frac{\sum_{i=2}^{n-2} |\epsilon_i|^p X_i}{n-3} \cdot \frac{n-3}{n+1} \left(\frac{n+1}{X} - 1\right) ,$$

and by the strong law of large numbers we have $\frac{\sum_{i=2}^{n-2} |\epsilon_i|^p X_i}{n-3} \overset{\text{a.s.}}{\longrightarrow} \mathbb{E} |\epsilon_i|^p X_i < \infty$ and $\left(\frac{n+1}{X}\right) \overset{\text{a.s.}}{\longrightarrow} 1$, and thus

$$\sum_{i=2}^{n-2} 4^{p-1} |\epsilon_i|^p \left(\ell_i - \tilde{\ell}_i\right) \overset{\text{a.s.}}{\longrightarrow} 0 . \quad (33)$$

By a similar argument, we also have

$$\sum_{i=2}^{n-2} 4^{p-1} |\epsilon_{i+1}|^p \left(\ell_i - \tilde{\ell}_i\right) \overset{\text{a.s.}}{\longrightarrow} 0 . \quad (34)$$

Combining equation 31, (32), (33) and (34), we get

$$\hat{R} - \tilde{R} = \sum_{i=2}^{n-2} 2^{3p-2} G^p \left( \ell_i^{p+1} - \tilde{\ell}_i^{p+1} \right) + \sum_{i=2}^{n-2} 4^{p-1} |\epsilon_i|^p \left( \ell_i - \tilde{\ell}_i \right) + \sum_{i=2}^{n-2} 4^{p-1} |\epsilon_{i+1}|^p \left( \ell_i - \tilde{\ell}_i \right)$$

$$+ \sum_{i=2}^{n-2} 4^{p-1} (|\epsilon_i| + |\epsilon_{i+1}| + |\epsilon_{i-1}| + |\epsilon_{i+2}|)^p \left( \frac{\ell_i^{p+1}}{\max\{\ell_{i-1}, \ell_{i+1}\}^p} - \frac{\tilde{\ell}_i^{p+1}}{\max\{\tilde{\ell}_{i-1}, \tilde{\ell}_{i+1}\}^p} \right) \xrightarrow{\text{a.s.}} 0$$

$\square$

*Proof of Claim D.3.* We first show that $\max\{A, B\} \stackrel{d}{=} A + \frac{B}{2}$. Both $A, B \stackrel{\text{i.i.d.}}{\sim} \text{Exp}(1)$. We will show that the CDF of $\max\{A, B\}$ is the same as the CDF of $A + \frac{B}{2}$. Let $F_{\max\{A,B\}}(.)$ and $F_{A+\frac{B}{2}}(.)$ be their CDFs respectively. Then for any $z \geq 0$.

$$F_{A+\frac{B}{2}}(z) = \mathbb{P}\left( A + \frac{B}{2} \leq z \right) = \int_0^z e^{-x} \mathbb{P}\left( B \leq 2(z-x) \right) \, dx = \int_0^z e^{-x} \left( 1 - e^{-2(z-x)} \right) \, dx$$

$$= \int_0^z e^{-x} \, dx - e^{-2z} \int_0^z e^x \, dx = 1 - e^{-z} - e^{-2z}(e^z - 1) = 1 - 2e^{-z} + e^{-2z}$$

$$= \left( 1 - e^{-z} \right)^2 = \mathbb{P}(A \leq z, B \leq z) = \mathbb{P}(\max\{A, B\} \leq z)$$

$$= F_{\max\{A,B\}}(z).$$

Using this, we can further simplify the expectation as follows.

$$\mathbb{E}\left[ \frac{1}{\max\{A, B\}^p} \right] = \mathbb{E}\left[ \frac{1}{(A + \frac{B}{2})^p} \right] \leq \mathbb{E}\left[ \frac{2^p}{(A+B)^p} \right] = 2^p \, \mathbb{E}\left[ \frac{1}{\Gamma(2, 1)^p} \right]$$

$$= 2^p \int_0^\infty \frac{1}{z^p} \cdot z e^{-z} \, dz$$

$$= 2^p \int_0^\infty z^{1-p} e^{-z} \, dz$$

$$= 2^p \Gamma(2 - p)$$

$$\leq \frac{2^p}{2 - p},$$

where the last inequality follows from Claim D.4 and the fact that $1 \leq p < 2$. $\square$

**Claim D.4.** *For $z \in (0, 1]$, we have $\frac{1}{2z} \leq \Gamma(z) \leq \frac{1}{z}$.*

*Proof.* For $z > 0$,

$$\Gamma(z) = \int_0^\infty w^{z-1} e^{-w} \, dw = \frac{w^z}{z} \cdot e^{-w} \Big|_0^\infty - \int_0^\infty -e^{-w} \frac{w^z}{z} \, dw = 0 + \frac{1}{z} \cdot \int_0^\infty e^{-w} w^z \, dw$$

$$= \frac{\Gamma(1 + z)}{z} \tag{35}$$

Now for $0 < z \leq 1$, we have $1 < 1 + z \leq 2$. Therefore,

$$\frac{1}{2} \leq \Gamma(1 + z) \leq 1.$$

The upper bound follows from the fact that $\Gamma(.)$ is unimodal (with first decreasing and then increasing). Therefore, the maximum value of $\Gamma(.)$ in $[1, 2]$ is $\max\{\Gamma(1), \Gamma(2)\} = 1$. The lower bound follows from Deming & Colcord (1935); the minimum value is approximately 0.8856032, which is at least $1/2$. Putting this back in equation 35, for $0 < z \leq 1$

$$\frac{1}{2z} \leq \Gamma(z) \leq \frac{1}{z}.$$

$\square$

# E    LOWER BOUNDS

Our lower bounds (Theorem 3 and 4) follow from very similar situations under which spikes are formed. Therefore, we first present the basic construction of such a situation; later, we specialize to the case of $p \in [1, 2)$ and $p \geq 2$ and obtain both theorems respectively.

Recall the noise model $\epsilon \sim \mathcal{N}(0, 1)$. We first claim the following.

**Lemma E.1.** *For any function $f$, we have $\mathcal{L}_p(f) \geq \mathcal{R}_p(f)$.*

The above lemma then allows us to solely focus on lower bounding the reconstruction loss of the predictor.

*Proof of Lemma E.1.* By definition,

$$\mathcal{L}_p(f) = \mathop{\mathbb{E}}_{(x,y)\sim\mathcal{D}} [|f(x) - y|^p] = \mathop{\mathbb{E}}_{x\sim\text{Uniform}([0,1]), \epsilon\sim\mathcal{N}(0,1)} [|f(x) - f^*(x) - \epsilon|^p].$$

Let $q : \mathbb{R}_+ \to \mathbb{R}_+$ be the density of the folded standard Gaussian, i.e. $|\mathcal{N}(0, 1)|$. Then due to the symmetry of the Gaussian,

$$\mathcal{L}_p(f) = \mathop{\mathbb{E}}_{x\sim\text{Uniform}([0,1])} \mathop{\mathbb{E}}_{\epsilon\sim\mathcal{N}(0,1)} [|f(x) - f^*(x) - \epsilon|^p]$$

$$= \mathop{\mathbb{E}}_{x\sim\text{Uniform}([0,1])} \left[ \int_0^\infty \left( \frac{1}{2}|f(x) - f^*(x) - \delta|^p + \frac{1}{2}|f(x) - f^*(x) + \delta|^p \right) q(\delta)\,d\delta \right]$$

$$\geq \mathop{\mathbb{E}}_{x\sim\text{Uniform}([0,1])} \left[ \int_0^\infty |f(x) - f^*(x)|^p q(\delta)\,d\delta \right] \qquad \text{(by Claim E.2 below)}$$

$$= \mathop{\mathbb{E}}_{x\sim\text{Uniform}([0,1])} [|f(x) - f^*(x)|^p] = \mathcal{R}_p(f). \qquad \text{(by definition)}$$

Here the second last step follows from the following claim, which we prove below.

**Claim E.2.** *For any $\mu \in \mathbb{R}$, $p \geq 1$, and $\delta \in \mathbb{R}_+$, we have $\frac{1}{2}(|\mu + \delta|^p + |\mu - \delta|^p) \geq |\mu|^p$.*

*Proof.* The claim trivially holds for $\mu = 0$. Now consider $\mu > 0$, then if $\delta \geq \mu$ then

$$\frac{1}{2}((\mu + \delta)^p + |\mu - \delta|^p) \geq \frac{1}{2}(2\mu)^p \geq 2^{p-1}\mu^p \geq \mu^p.$$

But if $\delta < \mu$ then since the function $|.|^p$ is convex for $p \geq 1$ we have

$$\frac{1}{2}(|\mu + \delta|^p + |\mu - \delta|^p) = \frac{1}{2}(\mu + \delta)^p + \frac{1}{2}(\mu - \delta)^p \geq \left( \frac{\mu + \delta + \mu - \delta}{2} \right)^p = \mu^p.$$

Finally, if $\mu < 0$, then apply the same argument to $-\mu$ giving the desired result. $\qquad \square$

This concludes the proof of the lemma. $\qquad \square$

We now describe the situations under which spikes are formed. We will consider $f^*(x) \equiv 0$ and the noise model $\epsilon \sim \mathcal{N}(0, 1)$. Let $S \sim \mathcal{D}^n$ be the data points. Again, we number them such that

$$x_0 =: 0 < x_1 < \cdots < x_n < 1 := x_{n+1}.$$

We have $n + 1$ intervals with lengths $\ell_0, \ldots, \ell_n$ from left to right. In particular, $\ell_i = x_{i+1} - x_i$.

We now consider $N := \lfloor n/10 \rfloor$ disjoint subsets $S_1, \ldots, S_N \subset S$ such that $S_i \cap S_j = \emptyset$ for $i \neq j$. Each $S_i$ is a set of six consecutive points (ordered from left to right). For any $i \in [N]$,

$$S_i := \{(x_j, y_j) : j = 10(i - 1) + 1, \ldots, 10(i - 1) + 6\}.$$

It is clear by definition that these sets are disjoint.

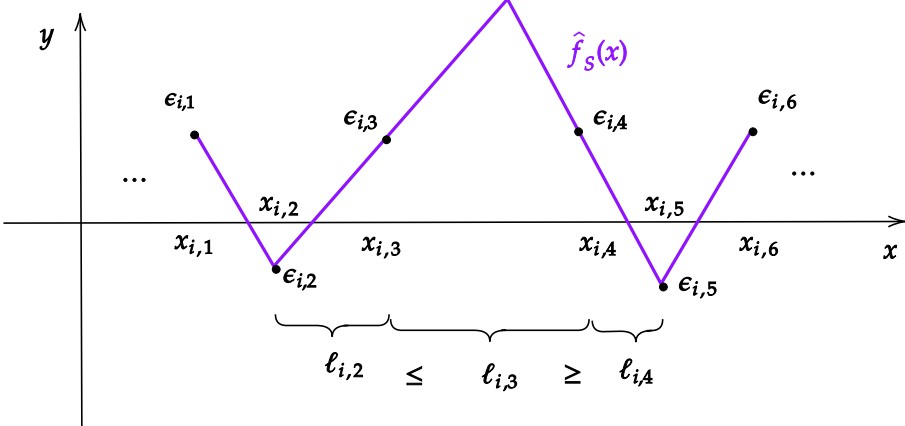

Figure 6: An illustration of configuration of points in the $i$-th subset $S_i$ when the event $A_i$ happens.

We denote the noise random variables associated with $j$-th point of $S_i$ by $\epsilon_{i,j}$. Also. we re-denote the distance between any two consecutive points from the same subset by $\ell_{i,j}$, i.e. $\ell_{i,j}$ is the distance between $(j+1)$-th point and $j$-th point in $S_i$. Consider the events $A_1, \ldots, A_N$ as below.

$$A_i := \{\epsilon_{i,2}, \epsilon_{i,5} \leq -1\} \cap \{\epsilon_{i,1}, \epsilon_{i,3}, \epsilon_{i,4}, \epsilon_{i,6} \in [1,2]\} \cap \{\ell_{i,3} \geq \ell_{i,2}\} \cap \{\ell_{i,3} \geq \ell_{i,4}\}$$

See Figure 6 for an illustration. This considers the event that the bad configuration of points happens for points in $S_i$. Then we can express $\mathcal{R}_p(\hat{f}_S)$, the risk of the predictor in $[0,1]$, in terms of the random variables $\ell_i$ s when such bad configurations occur. Formally,

**Lemma E.3.**    • *For $1 \leq p < 2$:*

$$\mathcal{R}_p(\hat{f}_S) \geq \sum_{i=1}^{N} \frac{\ell_{i,3}^{p+1}}{(\ell_{i,2} + \ell_{i,4})^p} \mathbb{1}[A_i];$$

• *For $p \geq 2$ :*

$$\mathcal{R}_p(\hat{f}_S) \geq \sum_{i=1}^{N} \frac{\ell_{i,3}^{3}}{(\ell_{i,2} + \ell_{i,4})^2} \mathbb{1}[A_i].$$

*Proof.* Without loss of generality, assume that $A_1$ holds and consider the following analysis under $A_1$. (One can choose any $i \in [N]$ and the discussion follows in the same way). If $A_1$ happens then $\mathsf{curv}(x_2) = +1$, $\mathsf{curv}(x_3) = \mathsf{curv}(x_4) = -1$ and again at $x_5$ we will have $\mathsf{curv}(x_5) = +1$. Therefore, $x_3$ and $x_5$ are special points according to Definition 4.3; the curvature changes within the distance of 2 points. Therefore, by Lemma 4.4, there is exactly one kink possible in $(x_2, x_5)$. More importantly, in the interval $(x_3, x_4)$, the function $\hat{f}_S(x) = \min\{g_2(x), g_4(x)\}$. Therefore, for any $x \in [x_3, x_4]$,

$$g_2(x) = y_3 + \left(\frac{y_3 - y_2}{x_3 - x_2}\right)(x - x_3) = \epsilon_3 + \frac{\epsilon_3 - \epsilon_2}{x_3 - x_2}(x - x_3) \geq 1 + \frac{2}{\ell_2}(x - x_3).$$

Similarly,

$$g_4(x) = y_4 + \frac{y_5 - y_4}{x_5 - x_4}(x - x_4) = \epsilon_4 + \frac{\epsilon_5 - \epsilon_4}{x_5 - x_4}(x - x_4) = \epsilon_4 + \frac{\epsilon_4 - \epsilon_5}{x_5 - x_4}(x_4 - x) \geq 1 + \frac{2}{\ell_4}(x_4 - x).$$

Therefore, if $A_1$ happened then writing the $L_p$ reconstruction error in the interval $[x_3, x_4]$:

$$
\begin{aligned}
\int_{x_3}^{x_4} |\hat{f}_S(x) - f^*(x)|^p dx &= \int_{x_3}^{x_4} |\hat{f}_S(x)|^p dx = \int_{x_3}^{x_4} \min\{g_2(x), g_4(x)\}^p \, dx \\
&\geq \int_{x_3}^{x_4} \min\{1 + \frac{2}{\ell_2}(x - x_3), 1 + \frac{2}{\ell_4}(x_4 - x)\}^p \, dx \\
&= \int_{x_3}^{x^*} \left(1 + \frac{2}{\ell_2}(x - x_3)\right)^p dx + \int_{x^*}^{x_4} \left(1 + \frac{2}{\ell_4}(x_4 - x)\right)^p \, dx \,,
\end{aligned}
$$

where $x^* \in [x_3, x_4]$ is the $x$-coordinate of the point of intersection of the two lines. Thus, to find $x^*$

$$
\begin{aligned}
1 + \frac{2}{\ell_2}(x^* - x_3) &= 1 + \frac{2}{\ell_4}(x_4 - x^*) \\
\frac{1}{\ell_2}(x^* - x_3) &= \frac{1}{\ell_4}(\ell_3 - (x^* - x_3)) \\
\left(\frac{1}{\ell_2} + \frac{1}{\ell_4}\right)(x^* - x_3) &= \frac{\ell_3}{\ell_4} \\
(x^* - x_3) &= \frac{\ell_3}{\ell_4} \cdot \frac{\ell_2\ell_4}{\ell_2 + \ell_4} = \frac{\ell_2\ell_3}{\ell_2 + \ell_4} \\
\implies (x_4 - x^*) &= \ell_3 - \frac{\ell_2\ell_3}{\ell_2 + \ell_4} = \frac{\ell_4\ell_3}{\ell_2 + \ell_4}
\end{aligned}
$$

**When $1 \leq p < 2$**: Writing the reconstruction risk in $[x_3, x_4]$ and substituting the above:

$$
\begin{aligned}
\int_{x_3}^{x_4} |\hat{f}_S(x) - f^*(x)|^p \, dx &\geq \int_{x_3}^{x^*} \left(1 + \frac{2}{\ell_2}(x - x_3)\right)^p dx + \int_{x^*}^{x_4} \left(1 + \frac{2}{\ell_4}(x_4 - x)\right)^p dx \\
&\geq \int_{x_3}^{x^*} \left(\frac{2}{\ell_2}(x - x_3)\right)^p dx + \int_{x^*}^{x_4} \left(\frac{2}{\ell_4}(x_4 - x)\right)^p dx \\
&= \frac{2^p}{(p+1) \cdot \ell_2^p}(x^* - x_3)^{p+1} + \frac{2^p}{(p+1) \cdot \ell_4^p}(x_4 - x^*)^{p+1} \\
&= \frac{2^p}{(p+1)} \left(\frac{\ell_2\ell_3^{p+1}}{(\ell_2 + \ell_4)^{p+1}} + \frac{\ell_4\ell_3^{p+1}}{(\ell_2 + \ell_4)^{p+1}}\right) \\
&= \frac{2^p}{(p+1)} \cdot \frac{\ell_3^{p+1}}{(\ell_2 + \ell_4)^p} \\
&\geq \frac{\ell_3^{p+1}}{(\ell_2 + \ell_4)^p},
\end{aligned}
\tag{36}
$$

where the last inequality follows from the fact that we are considering $p \in [1, 2)$.

**When p ≥ 2**

$$\int_{x_3}^{x_4} |\hat{f}_S(x) - f^*(x)|^p \, dx \geq \int_{x_3}^{x^*} \left(1 + \frac{2}{\ell_2}(x - x_3)\right)^p dx + \int_{x_*}^{x_4} \left(1 + \frac{2}{\ell_4}(x_4 - x)\right)^p dx$$

$$\geq \int_{x_3}^{x^*} \left(1 + \frac{2}{\ell_2}(x - x_3)\right)^2 dx + \int_{x^*}^{x_4} \left(1 + \frac{2}{\ell_4}(x_4 - x)\right)^2 dx$$

(both integrands are at least 1 in the range)

$$\geq \int_{x_3}^{x^*} \left(\frac{2}{\ell_2}(x - x_3)\right)^2 dx + \int_{x^*}^{x_4} \left(\frac{2}{\ell_4}(x_4 - x)\right)^2 dx$$

$$= \frac{4}{3\,\ell_2^2}(x^* - x_3)^3 + \frac{4}{3\,\ell_4^2}(x_4 - x^*)^3$$

$$= \frac{4}{3} \cdot \frac{\ell_2 \ell_3^3}{(\ell_2 + \ell_4)^3} + \frac{4}{3} \cdot \frac{\ell_4 \ell_3^3}{(\ell_2 + \ell_4)^3}$$

$$= \frac{4}{3} \cdot \frac{\ell_3^3}{(\ell_2 + \ell_4)^2}$$

$$\geq \frac{\ell_3^3}{(\ell_2 + \ell_4)^2}. \tag{37}$$

By the definition of risk, one can say that

$$\mathcal{R}_p(\hat{f}_S) \geq \sum_{i \in N} \int_{x_{i,3}}^{x_{i,4}} |\hat{f}_S(x) - f^*(x)|^p \, dx$$

Then by doing the above analysis as in equation 36 and equation 37 for each $i \in N$, along with the observation that the risk in any $[x_{i,3}, x_{i,4}]$ is non-negative, we obtain the desired result. In particular, for $1 \leq p < 2$ :

$$\mathcal{R}_p(\hat{f}_S) \geq \sum_{i=1}^{N} \frac{\ell_{i,3}^{p+1}}{(\ell_{i,2} + \ell_{i,4})^p} \mathbb{1}[A_i];$$

And, for $p \geq 2$ :

$$\mathcal{R}_p(\hat{f}_S) \geq \sum_{i=1}^{N} \frac{\ell_{i,3}^3}{(\ell_{i,2} + \ell_{i,4})^2} \mathbb{1}[A_i].$$

$\square$

**Lemma E.4.** *The events $\{A_i\}_{i=1}^N$ are independent. Moreover, $\mathbb{P}(A_i) = c_0$ for some universal constant $c_0$.*

*Proof.* The events $A_1, \ldots, A_N$ are independent since the noises are i.i.d. Also, the event about the interval lengths (that are dependent random variables) in the above $\{\ell_{i,3} \geq \ell_{i,2}\} \cap \{\ell_{i,3} \geq \ell_{i,4}\}$ can be expressed in terms of the underlying exponential random variables (which are independent) as $\{X_{i,3} \geq X_{i,2}\} \cap \{X_{i,3} \geq X_{i,4}\}$. Finally, it is easy to observe that for any $i \in [N]$, $\mathbb{P}(A_i)$ is some universal constant $c_0$. $\square$

### E.1 PROOF OF THEOREM 3

*Proof of Theorem 3.* We have $f^* \equiv 0$ and the noise distribution is $\mathcal{N}(0, 1)$. Therefore, $\mathcal{L}_p(f^*)$ is just a constant (dependent on $p$). More precisely,

$$L_p(f^*) = \mathbb{E}[|\epsilon|^p] = \sigma^p \cdot \frac{2^{p/2}}{\sqrt{\pi}} \Gamma\left(\frac{p+1}{2}\right) \leq \frac{2}{\sqrt{\pi}}, \tag{38}$$

where in the last inequality we use $\sigma = 1$ and $1 \leq p < 2$. Also, for $1 \leq p < 2$, by Lemma E.3

$$\mathcal{R}_p(\hat{f}_S) \geq \left(\sum_{i=1}^{N} \frac{\ell_{i,3}^{p+1}}{(\ell_{i,2} + \ell_{i,4})^p} \mathbb{1}[A_i]\right) := \hat{R}_p \tag{39}$$

Then, to create independence, we define $(\tilde{\ell}_0, \ldots, \tilde{\ell}_n) \overset{\text{i.i.d.}}{\sim} \text{Exp}(1)/(n+1)$ where $\tilde{\ell}_i = \frac{X_i}{n+1}$. We now replace $\ell_i$'s with $\tilde{\ell}_i$'s to define $\tilde{R}_p$.

$$\tilde{R}_p := \sum_{i=1}^{N} \frac{\tilde{\ell}_{i,3}^{p+1}}{(\tilde{\ell}_{i,2} + \tilde{\ell}_{i,4})^p} \mathbb{1}[A_i]$$

**Lemma E.5.** *As $n \to \infty$ we have $\hat{R}_p - \tilde{R}_p \overset{\text{a.s.}}{\longrightarrow} 0$.*

Moreover, expressing $\tilde{R}_p$ in terms of $X_i$'s

$$\tilde{R}_p = \frac{1}{(n+1)} \sum_{i=1}^{N} \frac{X_{i,3}^{p+1}}{(X_{i,2} + X_{i,4})^p} \mathbb{1}[A_i] = \frac{N}{(n+1)} \cdot \frac{1}{N} \sum_{i=1}^{N} \frac{X_{i,3}^{p+1}}{(X_{i,2} + X_{i,4})^p} \mathbb{1}[A_i] \quad (40)$$

Therefore, $\tilde{R}_p$ is the average of $N$ i.i.d. random variables (up to scaling). Each random variables have the expectation

$$\mathbb{E}\left[\frac{X_{i,3}^{p+1}}{(X_{i,2} + X_{i,4})^p} \mathbb{1}[A_i]\right] \leq \mathbb{E}\left[\frac{X_{i,3}^{p+1}}{(X_{i,2} + X_{i,4})^p}\right] = \mathbb{E}[X_{i,3}^{p+1}] \, \mathbb{E}\left[\frac{1}{(X_{i,2} + X_{i,4})^p}\right]$$

$$= \mathbb{E}[\text{Exp}(1)^{p+1}] \cdot \mathbb{E}\left[\frac{1}{\Gamma(2,1)^p}\right] \quad \text{(sum of two i.i.d. Exp(1)s is } \Gamma(2,1))$$

$$= \left(\int_0^\infty z^{p+1} e^{-z} \, dz\right) \left(\int_0^\infty \frac{1}{z^p} \cdot \frac{1}{\Gamma(2)} z e^{-z} \, dz\right)$$

$$= \left(\int_0^\infty z^{p+1} e^{-z} \, dz\right) \left(\int_0^\infty z^{1-p} e^{-z} \, dz\right)$$

$$= \Gamma(p+2)\Gamma(2-p) < \infty, \quad (41)$$

for $1 \leq p < 2$. Thus, by the strong law of large numbers, as $n \to \infty$ (also $N = \lfloor n/10 \rfloor \to \infty$)

$$\frac{1}{N} \sum_{i=1}^{N} \frac{X_{i,3}^{p+1}}{(X_{i,2} + X_{i,4})^p} \mathbb{1}[A_i] \overset{\text{a.s.}}{\longrightarrow} \mathbb{E}\left[\frac{X_{i,3}^{p+1}}{(X_{i,2} + X_{i,4})^p} \mathbb{1}[A_i]\right]$$

$$= \mathbb{P}(A_i) \, \mathbb{E}\left[\frac{X_{i,3}^{p+1}}{(X_{i,2} + X_{i,4})^p} \mid A_i\right] = \mathbb{P}(A_i) \, \mathbb{E}\left[\frac{X_{i,3}^{p+1}}{(X_{i,2} + X_{i,4})^p} \mid X_{i,2} \leq X_{i,3}, X_{i,4} \leq X_{i,3}\right]$$

$$\text{(by definition of } A_i)$$

$$\geq \mathbb{P}(A_i) \, \mathbb{E}\left[\frac{X_{i,3}^{p+1}}{(X_{i,2} + X_{i,4})^p}\right] \quad \text{(since the function is non-increasing in terms of } X_{i,2} \text{ and } X_{i,4})$$

$$= c_0 \, \mathbb{E}[\text{Exp}(1)^{p+1}] \cdot \mathbb{E}\left[\frac{1}{\Gamma(2,1)^p}\right] = c_0 \cdot \Gamma(2+p) \cdot \Gamma(2-p) \geq 2c_0 \cdot \Gamma(2-p)$$

$$\text{(since } 1 \leq p < 2)$$

$$\geq \frac{c_0}{(2-p)} \quad \text{(by Claim D.4)}$$

Moreover, $\lim_{n\to\infty} \frac{N}{n+1} = \lim_{n\to\infty} \frac{\lfloor n/10 \rfloor}{n+1} = \frac{1}{10}$. Therefore, recalling the definition of $\tilde{R}_p$ in equation 40, we obtain that for a constant $\gamma > 0$

$$\lim_{n\to\infty} \mathbb{P}\left[\tilde{R}_p \geq \frac{c_0}{10(2-p)} - \gamma\right] = 1.$$

Therefore, using Lemma E.5

$$\lim_{n\to\infty} \mathbb{P}\left[\hat{R}_p \geq \frac{c_0}{20(2-p)}\right] = 1.$$

Recalling equation 39 that $\hat{R}_p$ is a lower-bound on the risk, we obtain

$$\lim_{n\to\infty} \mathbb{P}\left[\mathcal{R}_p(\hat{f}_S) \geq \frac{c_0}{20(2-p)}\right] = 1.$$

By equation 38, $\mathcal{L}_p(f^*) \leq 2/\sqrt{\pi}$. Choosing $c := c_0\sqrt{\pi}/40$ we get the desired theorem, i.e.

$$\lim_{n \to \infty} \mathbb{P}_S\left[\mathcal{R}_p(\hat{f}_S) \geq \frac{c}{(2-p)}\mathcal{L}_p(f^*)\right] = 1.$$

Finally, using Lemma E.1 we obtain

$$\lim_{n \to \infty} \mathbb{P}_S\left[\mathcal{L}_p(\hat{f}_S) \geq \frac{c}{(2-p)}\mathcal{L}_p(f^*)\right] = 1.$$

$\square$

*Proof of Lemma E.5.* Recall equation 9 that $\ell_{i,j} \sim X_{i,j}/X$ for $1 \leq i \leq N$ and $j \in \{1, 2, 3, 4, 5, 6\}$, where $X_{i,j} \sim \mathrm{Exp}(1)$ and $X$ is the sum of $(n+1)$ i.i.d $\mathrm{Exp}(1)$ random variables. Therefore, by definition of $\hat{R}_p$ and $\tilde{R}_p$

$$\hat{R}_p - \tilde{R}_p = \left(\sum_{i=1}^{N} \frac{\ell_{i,3}^{p+1}}{(\ell_{i,2} + \ell_{i,4})^p}\mathbb{1}[A_i]\right) - \left(\sum_{i=1}^{N} \frac{\tilde{\ell}_{i,3}^{p+1}}{(\tilde{\ell}_{i,2} + \tilde{\ell}_{i,4})^p}\mathbb{1}[A_i]\right)$$

$$= \sum_{i=1}^{N}\left(\frac{(X_{i,3}/X)^{p+1}}{(X_{i,2}/X + X_{i,4}/X)^p}\mathbb{1}[A_i] - \frac{(X_{i,3}/n+1)^{p+1}}{(X_{i,2}/n+1 + X_{i,4}/n+1)^p}\mathbb{1}[A_i]\right)$$

$$= \underbrace{\frac{1}{n+1}\sum_{i=1}^{N}\left[\frac{X_{i,3}^{p+1}}{(X_{i,2} + X_{i,4})^p}\mathbb{1}[A_i]\right]}_{:=T}\left(\frac{n+1}{X} - 1\right) \tag{42}$$

We already know that by equation 41, $T$ converges to a finite quantity almost surely as $n \to \infty$. More formally, by the strong LLN

$$T = \frac{N}{n+1} \cdot \frac{1}{N}\sum_{i=1}^{N}\left[\frac{X_{i,3}^{p+1}}{(X_{i,2} + X_{i,4})^p}\mathbb{1}[A_i]\right] \xrightarrow{\text{a.s.}} \frac{1}{10} \cdot \mathbb{E}\left[\frac{X_{i,3}^{p+1}}{(X_{i,2} + X_{i,4})^p}\mathbb{1}[A_i]\right] < \infty,$$

where the last inequality follows from equation 41. Since $X$ is the sum of $(n+1)$ i.i.d. $\mathrm{Exp}(1)$ random variables, by the strong law of large numbers, as $n \to \infty$, we have $\frac{n+1}{X} \xrightarrow{\text{a.s.}} 1$. Therefore, $\left(\frac{n+1}{X} - 1\right) \xrightarrow{\text{a.s.}} 0$. Combining this in equation 42, we obtain that $\hat{R}_p - \tilde{R}_p \xrightarrow{\text{a.s}} 0$. $\square$

### E.2 PROOF OF THEOREM 4

*Proof of Theorem 4.* We have the same $f^* \equiv 0$ and the noise distribution is $\mathcal{N}(0, 1)$. By Lemma E.3 for $p \geq 2$

$$\mathcal{R}_p(\hat{f}_S) \geq \sum_{i=1}^{N} \frac{\ell_{i,3}^3}{(\ell_{i,2} + \ell_{i,4})^2} \cdot \mathbb{1}[A_i].$$

Recall equation 9 that $\ell_{i,j} \sim X_{i,j}/X$, where $X_{i,j} \sim \mathrm{Exp}(1)$ and $X$ is the sum of $(n+1)$ i.i.d $\mathrm{Exp}(1)$ random variables, one of which is $X_{i,j}$. Thus,

$$\mathcal{R}_p(\hat{f}_S) \geq \sum_{i=1}^{N} \frac{(X_{i,3}/X)^3}{(X_{i,2}/X + X_{i,4}/X)^2} \cdot \mathbb{1}[A_i] = \frac{1}{X}\sum_{i=1}^{N} \frac{X_{i,3}^3}{(X_{i,2} + X_{i,4})^2}\mathbb{1}[A_i] := \hat{R}. \tag{43}$$

For $i \in [N]$, define

$$\hat{R}_i := \frac{X_{i,3}^3\mathbb{1}[A_i]}{(X_{i,2} + X_{i,4})^2}. \quad \text{Then } \hat{R} = \frac{1}{X}\sum_{i=1}^{N}\hat{R}_i \tag{44}$$

Each $\hat{R}_i$ has an infinite expectation, as we show in the following calculation.

$$\mathbb{E}[\hat{R}_i] = \mathbb{E}\left[\frac{X_{i,3}^3 \mathbb{1}[A_i]}{(X_{i,2} + X_{i,4})^2}\right] = \mathbb{P}(A_i) \cdot \mathbb{E}\left[\frac{X_{i,3}^3}{(X_{i,2} + X_{i,4})^2} \mid A_i\right]$$

$$= c_0 \cdot \mathbb{E}\left[\frac{X_{i,3}^3}{(X_{i,2} + X_{i,4})^2} \mid X_{i,2} \leq X_{i,3}, X_{i,4} \leq X_{i,3}\right] \quad \text{(by Lemma E.4 and recall } A_i)$$

$$\geq c_0 \cdot \mathbb{E}\left[\frac{X_{i,3}^3}{(X_{i,2} + X_{i,4})^2}\right]$$

(the function inside expectation is non-increasing in terms of $X_{i,2}, X_{i,4}$)

$$= c_0 \cdot \mathbb{E}[\text{Exp}(1)^3] \cdot \mathbb{E}\left[\frac{1}{\Gamma(2,1)^2}\right]$$

$$= 6c_0 \cdot \int_0^\infty \frac{1}{z^2} \cdot z e^{-z} \, dz = 6c_0 \cdot \int_0^\infty \frac{e^{-z}}{z} \, dz = \infty.$$

Therefore, we define the truncated random variables $\hat{T}_i(a)$ which is the truncation of $\hat{R}_i$ at any finite threshold $a$. Formally,

$$\hat{T}_i(a) := \min\{\hat{R}_i, a\}.$$

Then $\hat{T}_i(a)$ has finite expectation by its definition, which we denote by $\mathbb{E}[\hat{T}_i(a)] = h(a)$ for some non-decreasing function $h(a)$. Also, it must be that $\lim_{a \to \infty} h(a) = \infty$, because the expectation of $\hat{T}_i(a)$ goes to infinity as $a$ does. We now define

$$\hat{T}(a) := \frac{1}{X} \sum_{i=1}^N \hat{T}_i(a),$$

then $\hat{T}(a)$ for any fixed $a$, by its definition serves as a lower bound on $\hat{R}$ (recall equation 44). We rewriting $\hat{T}(a)$ as follows

$$\hat{T}(a) := \frac{n+1}{X} \cdot \frac{N}{n+1} \cdot \frac{1}{N} \sum_{i=1}^N \hat{T}_i(a). \tag{45}$$

Then as $n \to \infty$, since $X$ is the sum of $n+1$ i.i.d. $\text{Exp}(1)$, by the strong law of large numbers

$$\frac{n+1}{X} \xrightarrow{\text{a.s.}} \mathbb{E}[\text{Exp}(1)] = 1. \tag{46}$$

Similarly, by the strong LLN

$$\frac{1}{N} \sum_{i=1}^N \hat{T}_i(a) \xrightarrow{\text{a.s.}} \mathbb{E}[\hat{T}_i(a)] = h(a). \tag{47}$$

And, finally using $\lim_{n \to \infty} \frac{N}{n+1} = \frac{1}{10}$ along with *equation* 47 and equation 46 in equation 45, we obtain

$$\hat{T}(a) \xrightarrow{\text{a.s.}} \frac{h(a)}{10}.$$

Therefore, for any $b > 0$ choose $a^*$ such that $h(a^*) \geq 21b$, then as $n \to \infty$, $\hat{T}(a^*) \xrightarrow{\text{a.s.}} h(a^*)/10 = 2.1b$ (note that it is always possible to choose such $a^*$ since $\lim_{a \to \infty} h(a) = \infty$). This implies

$$\lim_{n \to \infty} \mathbb{P}_S[\hat{T}(a^*) \geq 2b] = 1.$$

Thus, using the fact that $\hat{T}(a^*)$ is a lower bound on $\hat{R}$

$$\lim_{n \to \infty} \mathbb{P}_S[\hat{R} \geq 2b] = 1.$$

Finally, recalling equation 43 that $\hat{R}$ is a lower bound on $\mathcal{R}_p(\hat{f}_S)$ we obtain the desired theorem.

$$\lim_{n \to \infty} \mathbb{P}_S[\mathcal{R}_p(\hat{f}_S) > b] = 1.$$

Note that Lemma E.1 immediately also implies

$$\lim_{n \to \infty} \mathbb{P}_S[\mathcal{L}_p(\hat{f}_S) > b] = 1.$$

$\square$

# F  PROOF OF THEOREM 5

*Proof of Theorem 5.* Let $f^*$ be $G$-Lipschitz. First, we observe that the distance between two consecutive $x_i$'s is $1/n$. The domain $[0, 1]$ is divided into $n$ intervals of length $1/n$. We again denote $x_0 = 0$ for notational convenience. Our goal is to analyze the population $L_p$ loss of min-norm interpolating solution $\hat{f}_S(x)$ defined in equation 2.

$$
\begin{aligned}
\mathcal{L}_p(\hat{f}_S) &= \mathop{\mathbb{E}}_{(x,y)\sim\mathcal{D}} \left[ |\hat{f}_S(x) - y|^p \right] \\
&= \mathop{\mathbb{E}}_{x\sim\text{Uniform}([0,1]),\epsilon} \left[ |\hat{f}_S(x) - f^*(x) + \epsilon|^p \right] \\
&\leq 2^{p-1} \mathop{\mathbb{E}}_{x\sim\text{Uniform}([0,1]),\epsilon} \left[ |\hat{f}_S(x) - f^*(x)|^p + |\epsilon|^p \right] \\
&= 2^{p-1} \left( \mathop{\mathbb{E}}_{x\sim\text{Uniform}([0,1])} \left[ |\hat{f}_S(x) - f^*(x)|^p \right] + \mathop{\mathbb{E}}_{\epsilon} \left[ |\epsilon|^p \right] \right) \\
&= 2^{p-1} (\mathcal{R}_p(\hat{f}_S) + \mathcal{L}_p(f^*))
\end{aligned}
\tag{48}
$$

Therefore, it suffices to bound $\mathcal{R}_p(\hat{f}_S)$. It can be expressed as the following.

$$
\mathcal{R}_p(\hat{f}_S) = \sum_{i=0}^{n-1} \left( \underbrace{\int_{x_i}^{x_{i+1}} |\hat{f}(x) - f^*(x)|\, dx}_{:=R_i} \right)
\tag{49}
$$

For $x \in [0, x_2]$, $\hat{f}_S(x) = g_1(x)$ by Lemma 4.2. Thus we have

$$
\begin{aligned}
|\hat{f}_S(x) - f^*(x)| &= |g_1(x) - f^*(x)| \\
&= \left| y_1 + \frac{(y_2 - y_1)}{x_2 - x_1}(x - x_1) - f^*(x) \right| \\
&= \left| f^*(x_1) + \epsilon_1 + \left( \frac{f^*(x_2) + \epsilon_2 - f^*(x_1) - \epsilon_1}{x_2 - x_1} \right)(x - x_1) - f^*(x) \right| \\
&\leq |f^*(x_1) - f^*(x)| + |\epsilon_1| + \frac{G|x_2 - x_1| + |\epsilon_2 - \epsilon_1|}{x_2 - x_1}|x - x_1| \\
&\leq G|x - x_1| + |\epsilon_1| + \left( \frac{G(x_2 - x_1) + |\epsilon_2| + |\epsilon_1|}{x_2 - x_1} \right)|x - x_1| \\
&\leq G \cdot \frac{1}{n} + |\epsilon_1| + \left( \frac{G \cdot \frac{1}{n} + |\epsilon_2| + |\epsilon_1|}{\frac{1}{n}} \right) \cdot \frac{1}{n} \\
&= \frac{2G}{n} + |\epsilon_2| + 2|\epsilon_1|.
\end{aligned}
$$

Using this we get,

$$
\begin{aligned}
R_0 + R_1 &\leq \int_0^{x_2} |\hat{f}_S(x) - f^*(x)|^p\, dx \leq \frac{2}{n} \left( \frac{2G}{n} + |\epsilon_2| + 2|\epsilon_1| \right)^p \\
&\leq 3^p \left( \left( \frac{2G}{n} \right)^p + |\epsilon_2|^p + 2^p|\epsilon_1|^p \right) \cdot \frac{2}{n}.
\end{aligned}
\tag{50}
$$

Using exactly the same steps, one can also bound the risk in the last interval $[x_{n-1}, x_n]$.

$$
R_{n-1} \leq \int_{x_{n-1}}^{x_n = 1} |\hat{f}_S(x) - f^*(x)|^p\, dx \leq 3^p \left( \left( \frac{2G}{n} \right)^p + |\epsilon_{n-1}|^p + 2^p|\epsilon_n|^p \right) \cdot \frac{1}{n}.
\tag{51}
$$

Fix any $i \in \{2, \ldots, n-2\}$, and consider $x \in [x_i, x_{i+1}]$. By Lemma C.2, we get:

$$
|\hat{f}(x) - f^*(x)| \leq \max\left\{ |g_i(x) - f^*(x)|, \min\{|g_{i+1}(x) - f^*(x)|, |g_{i-1}(x) - f^*(x)|\} \right\},
$$

We can simplify each one by one.

$$
\begin{aligned}
|g_i(x) - f^*(x)| &= \left| y_i + \frac{(y_{i+1} - y_i)}{x_{i+1} - x_i}(x - x_i) - f^*(x) \right| \\
&= \left| f^*(x_i) + \epsilon_i + \frac{f^*(x_{i+1}) + \epsilon_{i+1} - f^*(x_i) - \epsilon_i}{x_{i+1} - x_i}(x - x_i) - f^*(x) \right| \\
&\leq |f^*(x_i) - f^*(x)| + |\epsilon_i| + \left( \frac{G/n + |\epsilon_{i+1}| + |\epsilon_i|}{1/n} \right) \cdot |(x - x_{i-1})| \\
&\leq \frac{G}{n} + |\epsilon_i| + \left( \frac{G/n + |\epsilon_{i+1}| + |\epsilon_i|}{1/n} \right) \cdot \frac{1}{n} \\
&= \frac{2G}{n} + 2|\epsilon_i| + |\epsilon_{i+1}|.
\end{aligned}
$$

Similarly,

$$
\begin{aligned}
|g_{i-1}(x) - f^*(x)| &= \left| y_i + \frac{(y_i - y_{i-1})}{x_i - x_{i-1}}(x - x_i) - f^*(x) \right| \\
&= \left| f^*(x_i) + \epsilon_i + \frac{f^*(x_i) + \epsilon_i - f^*(x_{i-1}) - \epsilon_{i-1}}{x_i - x_{i-1}}(x - x_i) - f^*(x) \right| \\
&\leq |f^*(x_i) - f^*(x)| + |\epsilon_i| + \left( \frac{G/n + |\epsilon_i| + |\epsilon_{i-1}|}{1/n} \right) |(x - x_i)| \\
&\leq \frac{G}{n} + |\epsilon_i| + \left( \frac{G/n + |\epsilon_i| + |\epsilon_{i-1}|}{1/n} \right) \cdot \frac{1}{n} \\
&= \frac{2G}{n} + 2|\epsilon_i| + |\epsilon_{i-1}|.
\end{aligned}
$$

Using a similar strategy, we also bound:

$$
\begin{aligned}
|g_{i+1}(x) - f^*(x)| &= \left| y_{i+1} + \frac{(y_{i+2} - y_{i+1})}{x_{i+2} - x_{i+1}}(x - x_{i+1}) - f^*(x) \right| \\
&= \left| f^*(x_{i+1}) + \epsilon_{i+1} + \frac{f^*(x_{i+2}) + \epsilon_{i+2} - f^*(x_{i+1}) - \epsilon_{i+1}}{x_{i+2} - x_{i+1}}(x - x_{i+1}) - f^*(x) \right| \\
&\leq |f^*(x_{i+1}) - f^*(x)| + |\epsilon_{i+1}| + \left( \frac{G/n + |\epsilon_{i+2}| + |\epsilon_{i+1}|}{1/n} \right) |(x - x_{i+1})| \\
&\leq \frac{G}{n} + |\epsilon_{i+1}| + \left( \frac{G/n + |\epsilon_{i+2}| + |\epsilon_{i+1}|}{1/n} \right) \cdot \frac{1}{n} \\
&= \frac{2G}{n} + 2|\epsilon_{i+1}| + |\epsilon_{i+2}|.
\end{aligned}
$$

Therefore, combining the above three, for $x$ in the intermediate intervals:

$$
\begin{aligned}
|\hat{f}(x) - f^*(x)| &\leq \max \left\{ \frac{2G}{n} + 2|\epsilon_i| + |\epsilon_{i+1}|, \frac{2G}{n} + 2|\epsilon_i| + |\epsilon_{i-1}|, \frac{2G}{n} + 2|\epsilon_{i+1}| + |\epsilon_{i+2}| \right\} \\
&\leq \frac{2G}{n} + 2|\epsilon_i| + 2|\epsilon_{i+1}| + |\epsilon_{i+2}| + |\epsilon_{i-1}|.
\end{aligned}
$$

Therefore, for any $i \in \{2, \dots, n-2\}$, and $x \in [x_i, x_{i+1}]$,

$$
|\hat{f}(x) - f^*(x)|^p \leq 5^p \left( \left( \frac{2G}{n} \right)^p + 2^p |\epsilon_i|^p + 2^p |\epsilon_{i+1}|^p + |\epsilon_{i+2}|^p + |\epsilon_{i-1}|^p \right)
$$

Therefore,

$$
R_i = \int_{x_i}^{x_{i+1}} |\hat{f}(x) - f^*(x)|^p \, dx \leq 5^p \left( \left( \frac{2G}{n} \right)^p + 2^p |\epsilon_i|^p + 2^p |\epsilon_{i+1}|^p + |\epsilon_{i+2}|^p + |\epsilon_{i-1}|^p \right) \frac{1}{n}.
$$

We now sum over all $i \in \{0, 2, \ldots, n-1\}$ and use equation 50 and equation 51:

$$\mathcal{R}_p(\hat{f}_S) = \sum_{i=0}^{n-1} R_i \leq \sum_{i=2}^{n-2} \left[ 5^p \left( \left( \frac{2G}{n} \right)^p + 2^p |\epsilon_i|^p + 2^p |\epsilon_{i+1}|^p + |\epsilon_{i+2}|^p + |\epsilon_{i-1}|^p \right) \frac{1}{n} \right]$$

$$+ 3^p \left( \left( \frac{2G}{n} \right)^p + |\epsilon_2|^p + 2^p |\epsilon_1|^p \right) \cdot \frac{2}{n} + 3^p \left( \left( \frac{2G}{n} \right)^p + |\epsilon_{n-1}|^p + 2^p |\epsilon_n|^p \right) \cdot \frac{1}{n}$$

$$\leq \frac{(5^p + 2 \cdot 3^p + 3^p)(2G)^p}{n^p} + \frac{2 \cdot 3^p |\epsilon_2|^p + 2 \cdot 6^p |\epsilon_1|^p + 3^p |\epsilon_{n-1}|^p + 6^p |\epsilon_n|^p}{n}$$

$$+ \frac{1}{n} \sum_{i=2}^{n-2} (10^p |\epsilon_i|^p + 10^p |\epsilon_{i+1}|^p + 5^p |\epsilon_{i+2}|^p + 5^p |\epsilon_{i-1}|^p)$$

$$\leq \frac{(5^p + 3^{p+1})(2G)^p}{n^p} + \frac{2 \cdot (10^p + 5^p)}{n} \cdot \sum_{i=1}^{n} |\epsilon_i|^p \tag{52}$$

The first term converges to 0 surely as $n \to \infty$. The second term is proportional to the average of $n$ i.i.d. random variables $|\epsilon_i|^p$, each having $\mathbb{E}[|\epsilon_i|^p] = \mathcal{L}_p(f^*)$. Therefore, by the strong LLN

$$\frac{1}{n} \sum_{i=1}^{n} |\epsilon_i|^p \xrightarrow{a.s.} \mathcal{L}_p(f^*).$$

Combining this with equation 52:

$$\lim_{n \to \infty} \mathop{\mathbb{P}}_{S} \left[ \mathcal{R}_p(\hat{f}_S) \leq (2 \cdot 10^p + 2 \cdot 5^p + 1) \mathcal{L}_p(f^*) \right] = 1.$$

Finally, substituting the bound from equation 48 in the above:

$$\lim_{n \to \infty} \mathop{\mathbb{P}}_{S} \left[ \mathcal{L}_p(\hat{f}_S) \leq (20^p + 10^p + 2^p) \mathcal{L}_p(f^*) \right] = 1.$$

Letting $C_p = 20^p + 10^p + 2^p$, we get the desired theorem. $\qquad \square$

