# OpenReview forum: "Noisy Interpolation Learning with Shallow Univariate ReLU Networks"
_ICLR.cc/2024/Conference — ICLR 2024 spotlight_

### Official Review · Reviewer_toQ6 · 2023-10-29

**Soundness:** 4 excellent
**Presentation:** 3 good
**Contribution:** 3 good
**Rating:** 8
**Confidence:** 4

**Summary:**

This paper studies the min-$\ell_2$ norm interpolation learning of two-layer ReLU neural networks under the univariate data. The results start from linear splines, demonstrating that linear-spline interpolator exhibits tempered behavior. Then the min-norm interpolator is studied based on the relationship with the linear-spline. Then the results show that, tempered overfitting occurs in the Lp space with $1 \leq p <2$ but catastrophic overfitting occurs for $p > 2$.

**Strengths:**

The generalization performance of linear splines and min-norm solution is studied in terms of tempered overfitting and catastrophic overfitting

**Weaknesses:**

There is no distinct drawback in my view. The proof is based on the nice statistical property of $\ell_i$, which might be the key technical difficulty when extending to the $d$-dimensional data. For example, the two-dimensional data, we sort the data, split the space, define the risk in the two-dimensional interval. But the estimation based on $\ell_i$ is unclear to me.

Besides, comparison with (Kornowski et al. 2023) requires more discussion, especially in terms of the technical tools.

Apart from this, some references on the following topics in the related work are missing:
-	Neural networks, the representer theorem, splines, Banach spaces
-	Benign overfitting papers

**Questions:**

- Could you please intuitively explain why catastrophic overfitting occurs for $p>2$ under the min-norm solution while linear spline does not?

- Can Linear splines obtain better performance than the min-norm solution?
- What is the result under the min-$l1$ norm solution? Intuitively, there is no significant difference for univariate data but I expect that some results from previous min-l1 norm literature can be discussed under the univariate setting.
- What’s the meaning of “mildly dependent random variables” of $\ell_i$?
- How does $(n+1)/X -> 1$ hold with almost surely?

---

> ### Author Response · Authors · 2023-11-21
> **Addressing Questions of Reviewer toQ6**
>
> We thank the reviewer for the positive feedback on the work! We will add a more detailed comparison with Kornowski et. al. [1] and a discussion on the topics raised by the reviewer. We address their primary questions in order:
>
> ------
> * **Intuition regarding catastrophic vs tempered overfitting for $p>2$:** It’s best to see this through Figure 2.  The value of the linear splines stays between the value of the observed labels, and so in a sense are not too “crazy” and their distance from the target is similar to the magnitude of the noise, hence the $L_p$ error (for any $p$) is proportional to that of $p$-th moment of the noise (the Bayes $L_p$ error), which corresponds to “*tempered*” overfitting (tempered here means excess error proportional to the Bayes error, or noise level). More technically, consider $f^* \equiv 0$ for simplicity. The fitted function in any interval $[x_i,x_{i+1}]$ linearly interpolates between $y_i$ and $y_{i+1}$ and does not exceed the label values. Moreover, these labels are just noise random variables $\varepsilon_i$ and $\varepsilon_{i+1}$ when $f^* \equiv 0$, thus, for linear splines the $L_p$ risk will be proportional to the $p$-th moment of noise random variables, which is the Bayes $L_p$ risk.
>
>     However, as illustrated in Figure 2, min norm neural nets prefer solutions with fewer “kinks”, which leads to spikes that go well
>     beyond the observed values.  With more points, there will be some extreme spikes, but these extreme spikes will also be *narrow*.
>     For higher $p$, the contribution of extreme spikes to the $L_p$ error is more significant (scaling as height$^p$), and $p=2$ is the tip-off point
>    where the thin and extreme spikes dominate the error. At a technical level, the height of the spike in the interval $[x_i,x_{i+1}]$ of length
>    $\ell_i$ depends on the relative ratio of $\ell_i$ and its neighboring intervals $\ell_{i-1}$ and $\ell_{i+1}$. Thus, if $\ell_{i-1}$ and $\ell_{i+1}$ are "very small” as compared to $\ell_{i}$ then the effect of spikes would be large. The precise dependence on $\ell_i$ s and $p$ in the risk comes up in eq 25, which is analyzed using the distributional properties of $\ell_i$ s. Note that when the input is on the grid and the points are equally spaced, the relative ratio of $\ell_{i}$ with $\ell_{i-1}$ and $\ell_{i+1}$ remains constant. Thus, the asymptotic risk is worsened but only by a constant factor from linear splines, and we get tempered overfitting for $L_p$ losses for any $p \geq 1$.
>
> * **Linear splines vs min-norm networks:** The paper indeed establishes that if we insist on overfitting, then min-norm networks are more problematic than linear splines, and we are better off with linear splines.  But even better would be to not overfit in the first place and balance norm with training error, in which case low-norm networks would be better (e.g. for Lipschitz functions + noise, balancing network norm with training error would converge to the Bayes optimal predictor, whereas linear splines would be *“tempered”* with population error larger than Bayes optimal).  Disclaimer: the purpose of this paper is NOT to advocate for overfitting, but rather to study its effect, in line with recent interest in the topic.
>
> * We are not aware of work looking at min-$\ell_1$-norm neural nets, and are not sure how they would behave, but this could be interesting to look at.
>
> * **“Mildly dependent”:** It is an informal term used to give intuition, not in a formal context.  It roughly means that although the random variables are not, strictly speaking, independent, the effect of the dependencies is small, and vanishes as $n→\infty$.  This is again an informal statement—-in our analysis, this dependence is taken care of precisely.
>
> * Note that $X$ is the sum of $(n+1)$ i.i.d. $\textnormal{Exponential}(1)$. Thus, by the strong law of large numbers, as $n→ \infty$, $X/(n+1) \rightarrow \mathbb{E}[\textnormal{Exponential}(1)]=1$ almost surely. By the laws of limits and the definition of almost surely convergence, this implies that $(n+1)/X \rightarrow 1$ almost surely.
>
> ------
> [1] Guy Kornowski, Gilad Yehudai, and Ohad Shamir. From tempered to benign overfitting in relu neural networks. arXiv preprint arXiv:2305.15141, 2023.

---

### Official Review · Reviewer_9Vzu · 2023-10-31

**Soundness:** 4 excellent
**Presentation:** 4 excellent
**Contribution:** 4 excellent
**Rating:** 8
**Confidence:** 4

**Summary:**

This paper delves into the nuanced behaviors of tempered overfitting and catastrophic overfitting within regression scenarios employing a min-norm two-layer ReLU network with skip connections. The key contribution lies in the establishment of significant results, notably demonstrating the occurrence of catastrophic overfitting when the $L_p$ loss is applied with $p\geq2$. Furthermore, the paper uncovers the phenomenon of tempered overfitting, which surfaces when utilizing the $L_p$ loss with $1\leq p<2$.

In a noteworthy extension of its findings, the paper also establishes that when working with samples distributed on a grid, tempered overfitting manifests for the $L_p$ loss with $p\geq1$. These results shed valuable light on the interplay between loss functions, network architecture, and data distribution in the context of regression with min-norm ReLU networks.

**Strengths:**

This paper is exceptionally well-crafted, boasting a highly organized structure that enhances its clarity and readability. The main paper is thoughtfully structured, and the presentation of the proof concept is remarkably accessible, thanks in part to the informative graphs provided.

The theoretical framework is excellent, as this paper conducts a comprehensive examination of the overfitting tendencies observed in min-norm ReLU networks within the context of regression. In doing so, it effectively bridges a critical gap, especially when contrasted with the closely related work by Kornowski et al. (2023), which primarily addressed overfitting within a classification setting.

**Weaknesses:**

It appears that this paper can be regarded as a subsequent work to Boursier & Flammarion (2023). The connection is evident as the pivotal lemma (Lemma 2.1) employed in this paper is directly drawn from Boursier & Flammarion (2023). Furthermore, the neural network model studied in this paper aligns with the one extensively examined in Boursier & Flammarion (2023). Consequently, the technical innovation in this paper seems somewhat limited in this regard.

It's important to note that this paper adopts a one-dimensional perspective, assuming $x$ to be a single dimension, confined within the range $x\sim[0,1]$. In contrast, Kornowski et al. (2023) considered high-dimensional $x$. Another differentiating factor is that this paper primarily focuses on characterizing the asymptotic behavior of population error and reconstruction error, while Kornowski et al. (2023) delved into an analysis that extends beyond the asymptotic realm.

In Theorem 4, which addresses catastrophic overfitting for $L_p$ with $p\geq 2$, it is essential to note that only a specific case with $f^*=0$ is considered. To enhance the depth and relevance of the analysis, it would be particularly intriguing to explore this phenomenon with a more general $f^*$.

**Questions:**

1. I have observed that in both your Theorem 1 and Theorem 2, you made the assumption that $f^*$ is a Lipschitz function. Assuming the Lipschitz constant is denoted as $c$, I am interested in understanding how the constants $C_p$ in Theorem 1 and the constant $C$ in Theorem 2 are related to this Lipschitz constant $c$.

2. What would be the impact on the analysis if we were to assume that $f^*$ is Holder continuous instead of Lipschitz continuous in your main theorem? Could this alternative assumption be beneficial, given that you are dealing with $L_p$ loss in this context?

---

> ### Author Response · Authors · 2023-11-20
> **Response to Reviewer 9Vzu**
>
> We thank the reviewer for their thorough review and positive feedback on the work!
>
> ------
>
> We want to first address some of the points mentioned as weaknesses, where we see things differently:
> * **Novelty relative to Boursier and Flammarion [1]:** Indeed, we study the same model as in Boursier and Flammarion [1].  This is also the same model studied in several other papers (e.g. Severese et. al. [2], Ergen and Pilanci [4], Hanin [5], and many more). The main novelty is in the question we ask, which is completely different: none of the previous papers, including Boursier and Flammarion [1], asked how such a model overfits, i.e. how it behaves when interpolating noisy data. In particular, our main results do not appear in any way in previous papers. To answer this novel question we indeed build on a characterization given by Boursier and Flammarion [1], but this is just a component of our analysis.
>
> * **Limiting to one dimension:** This is indeed a limitation, which is shared also by many other papers studying low-norm ReLU networks (e.g. Boursier and Flammarion [1], Hanin [5], Severevse et. al. [2], Debarre et. al. [3] and others mentioned above) and in-fact also the Kornowski et. al. [6] paper mentioned.  Kornowski et. al.’s treatment of higher dimensions is limited to simulations or the extreme case $d \gg n^2$ (in which case learning is essentially linear and the overfitting is benign—see paragraph 2 on page 1 and also the footnote)--they also do not have theoretical analysis for *fixed* dimensions beyond $d=1$.  Going beyond one dimension is very interesting, and we believe it is possible, but has proven to be difficult not only for us but also for the entire community.
>
> * **Considering** $f^* \equiv 0$ **for catastrophic overfitting:** As this is a lower bound, we find it strongest to show that overfitting is catastrophic even in the easiest case, i.e. $f^* \equiv 0$. Our result should be interpreted as: "Even for such a simple target function $f^* \equiv 0$, in the presence of noise, the interpolating predictor exhibits catastrophic behavior."  Overfitting will also be catastrophic with any Lipschitz target and non-zero noise.
>
> ----
> Below we answer their questions.
>
> * Our constant $C_p$ (in Theorem 1) only depends on $p$ and $C$ (in Theorem 2) is a universal constant (the dependence on $p$ is explicit). These constants do not have any dependence on the Lipschitz constant. Our result should be interpreted as “as long as the Lipschitz constant is finite, the asymptotic $L_p$ risk is proportional to the Bayes $L_p$ risk, where the constant of proportionality only depends on $p$.” Note that the dependence in the Lipschitz constant will show up in the non-asymptotic rates, but at least for the asymptotic results as we show here, the terms associated with the Lipschitz constant vanish as $n→ \infty$, and we get no dependence.
>
> * We believe our upper bounds of tempered overfitting continue to hold for any $(G,\alpha)$ Holder continuous functions with $0<\alpha \leq 1$. The same phenomenon of transition from tempered to catastrophic at $p=2$ would occur. The primary reason we consider Lipschitz continuity is because the Lipschitz functions can be exactly represented by ReLU nets with bounded weights (see Boursier and Flammarion [1], Theorem 1). If the function is not Lipschitz, then its representation cost (minimum norm to represent the function exactly) is infinite. However, we agree that even if one cannot represent the function *exactly*, it is possible to talk about *approximation*, and it seems like the same analysis generalizes to even Holder continuous functions. The only difference is that now the terms with dependence on $G$ vanish but at a slower rate.
>
> -----
> [1] Etienne Boursier and Nicolas Flammarion. Penalizing the biases in norm regularisation enforces sparsity. arXiv preprint arXiv:2303.01353, 2023.
>
> [2] Pedro Savarese, Itay Evron, Daniel Soudry, and Nathan Srebro. How do infinite-width bounded norm networks look in function space? In Conference on Learning Theory, pp. 2667–2690. PMLR, 2019.
>
> [3] Thomas Debarre, Quentin Denoyelle, Michael Unser, and Julien Fageot. Sparsest piecewise-linear regression of one-dimensional data. Journal of Computational and Applied Mathematics, 406:114044, 2022.
>
> [4] Tolga Ergen and Mert Pilanci. Convex geometry and duality of over-parameterized neural networks. Journal of Machine Learning Research, 2021.
>
> [5] Boris Hanin. Ridgeless interpolation with shallow relu networks in 1d is nearest neighbor curvature extrapolation and provably generalizes on Lipschitz functions. arXiv preprint arXiv:2109.12960, 2021.
>
> [6] Guy Kornowski, Gilad Yehudai, and Ohad Shamir. From tempered to benign overfitting in relu neural networks. arXiv preprint arXiv:2305.15141, 2023.

---

### Official Review · Reviewer_bbgZ · 2023-10-31

**Soundness:** 3 good
**Presentation:** 3 good
**Contribution:** 3 good
**Rating:** 8
**Confidence:** 4

**Summary:**

This paper tries to understand the generalization performance of overparametrized neural networks when interpolating noisy training data. Specifically, the authors consider the univariate 2-layer ReLU networks in regression setting and focus on the min $\ell_2$ norm interpolator. This paper shows that the generalization performance is subtle and depends on the factors such as the choice of loss at evaluation time and whether one is considering high probability case or taking expectation over the training samples. The overfitting is tempered (test loss neither goes to Bayes optimal nor to infinity) when loss is $L_p$ for $1<p<2$ and considering the high probability outcomes. The overfitting is catastrophic (test loss goes to infinity) when loss is $L_2$ or considering expectations.

**Strengths:**

-	Understanding the generalization performance of interpolating solutions, especially non-linear interpolators such as ReLU networks, is an important and interesting question in deep learning theory.
-	The paper gives a detailed characterization of the min $\ell_2$ norm interpolating ReLU networks, compares it with the linear splines, and shows the subtle generalization performance depending on the loss function. I believe this is a good result and it seems to be novel in the literature of implicit bias and benign overfitting.
-	The paper is overall well-written and easy-to-follow.

**Weaknesses:**

-	The paper focuses on the univariate ReLU networks which is relatively simple. (though it is understandable from technical point of view)
-	The results are in the asymptotic regime that sample size $n$ goes to infinity. Thus, it does not give explicit rate of convergence.

**Questions:**

-	The results in this paper seem to consider the asymptotic regime that sample size $n$ goes to infinity. I was wondering if there is a rate for this asymptotic convergence.
-	I was wondering if the min-norm interpolation considered in the paper can be naturally reached based on the implicit bias of some simple algorithms. For example, the authors mentioned that Shevchenko et al. showed similar spikes as the current paper. Does their algorithm exactly lead to the same implicit bias here?
-	I wonder if results like Theorem 1 and Theorem 5 are enough to say they are tempered overfitting, as they are only upper bound on the test loss. I feel there needs to also have a lower bound on the test loss?

---

> ### Author Response · Authors · 2023-11-20
> **Adressing the questions of Reviewer bbgZ**
>
> We seriously thank the reviewer for their constructive feedback! Below we address the questions.
>
> * In line with other papers on interpolation learning, we indeed provide an asymptotic characterization (benign/tempered/catastrophic).  Obtaining a finite sample bound on the convergence (or divergence) would require additional assumptions (which we preferred not to get into in order to keep the paper focused—e.g. our results do not depend on the Lipschitz constant, only on it being finite) but should be possible using standard concentration inequalities and a more elaborate analysis.
>
> * As we explain on page 2, the precise implicit bias of GD (and even Shevchenko et. al. [1]’s analysis) does not guarantee that min-norm interpolators are reached. However, we note that the main property of min-norm interpolators that determines the overfitting behavior in our analysis is the formation of “spikes”, and such spikes are empirically observed while interpolation with GD in Shevchenko et. al. [1]. Thus, studying min-norm interpolators with these properties is a good starting point. Moreover, considering the min-norm interpolator is natural when training with small weight decay.
>
> * Indeed, for linear splines (warmup) and data on the grid (contrast analysis), we do not explicitly provide lower bounds showing that overfitting is tempered rather than benign.  These are fairly straightforward and we will include them in the final version for completeness. Thanks for pointing this out! To be clear: we do provide lower bounds in the main analysis (min-norm neural nets on i.i.d. samples), just not for the warmup and the contrast analysis.
> -----
> [1] Alexander Shevchenko, Vyacheslav Kungurtsev, and Marco Mondelli. Mean-field analysis of piecewise linear solutions for wide relu networks. The Journal of Machine Learning Research, 23(1): 5660–5714, 2022.

---

> > ### Comment · Reviewer_bbgZ · 2023-11-23
> >
> > Thanks for the responses. I will keep my score.

---

### Meta-Review · Area_Chair_g31p · 2023-12-21

**Metareview:**

Based on the comprehensive reviews provided by the three reviewers (bbgZ, 9Vzu, and toQ6), I recommend the acceptance of Submission774 for publication. The reviewers have consistently identified the paper as making a significant contribution to the understanding of generalization performance in overparametrized neural networks, particularly focusing on the nuanced behaviors of tempered and catastrophic overfitting within regression scenarios using min-norm two-layer ReLU networks.

The reviewers have commended the paper for its clear presentation, well-structured arguments, and the depth of its theoretical analysis. The insights into how the choice of loss function, network architecture, and data distribution affect the model's generalization behavior are deemed particularly valuable. The paper is praised for bridging a critical gap in the literature and for its potential to inform future research on the implicit biases of neural networks and their relationship with overfitting.

The reviewers' concerns and questions have been adequately addressed by the authors, providing further clarification on the assumptions and implications of their results. The authors' responses have helped to reinforce the significance of the paper's findings, and the clarifications regarding the nuances of the model and its behavior under different conditions have been insightful.

Overall, the paper appears to be a strong contribution to the field of machine learning theory, and its acceptance seems warranted given the positive feedback from knowledgeable reviewers.

**Justification For Why Not Higher Score:**

The paper was not given a higher score due to its focus on univariate ReLU networks, which, as noted by Reviewer 9Vzu, is a limitation when compared to studies that consider high-dimensional data. Additionally, while the paper builds on previous work, particularly the model from Boursier & Flammarion (2023), there is a question of technical novelty that prevents a perfect score.

**Justification For Why Not Lower Score:**

Conversely, the paper was not given a lower score because it successfully addresses a novel question regarding how such models overfit when interpolating noisy data. The paper's results are substantial and do not appear to have been previously reported, indicating that the contribution is both novel and significant within its scope. Furthermore, the authors have demonstrated a clear understanding of the limitations of their work and have provided a roadmap for future research to expand upon their findings.

---

### Decision · Program_Chairs · 2024-01-16

Accept (spotlight)